# Synthetic Face Datasets Generation via Latent Space Exploration from Brownian Identity Diffusion

**David Geissbühler** [* 1]  **Hatef Otroshi Shahreza** [* 1 2]  **Sébastien Marcel** [1 3]

## Abstract

Face recognition models are trained on large-scale datasets, which have privacy and ethical concerns. Lately, the use of synthetic data to complement or replace genuine data for the training of face recognition models has been proposed. While promising results have been obtained, it still remains unclear if generative models can yield diverse enough data for such tasks. In this work, we introduce a new method, inspired by the physical motion of soft particles subjected to stochastic Brownian forces, allowing us to sample identities distributions in a latent space under various constraints. We introduce three complementary algorithms, called Langevin, Dispersion, and DisCo, aimed at generating large synthetic face datasets. With this in hands, we generate several face datasets and benchmark them by training face recognition models, showing that data generated with our method exceeds the performance of previously GAN-based datasets and achieves competitive performance with state-of-the-art diffusion-based synthetic datasets. While diffusion models are shown to memorize training data, we prevent leakage in our new synthetic datasets, paving the way for more responsible synthetic datasets. Project page: https://www.idiap.ch/paper/synthetics-disco

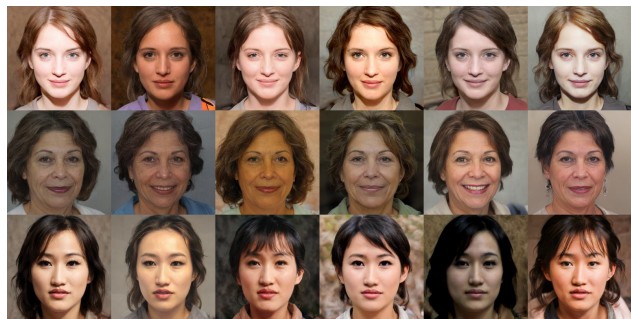

Figure 1: Example of synthetic faces generated using StyleGAN2. The three rows are three different classes generated with the *Langevin* algorithm while the columns show *intra-class* variations generated using the *Dispersion* algorithm.

large biometric datasets that have been collected *in the wild*, for instance by scraping images from the internet. Moreover, in addition to privacy concerns, biometric data might give a biased representation of the population depending on the data collection procedure (Drozdowski et al., 2020). Recently, MS-Celeb (Guo et al., 2016), a very popular dataset commonly used to train face recognition (FR) models, was withdrawn after exposure in the media of its privacy, fairness and demographic biases problems. While data collection campaigns performed in laboratories can be made representative of the general demographics and performed with subjects consents, they are typically quite limited due to the large amount of effort they require.

In this work, we propose another step towards the tackling of some of theses issues by developing physics-inspired algorithms that allows precise control on the sampling of synthetic identities and variations thereof. More precisely, our method treats samples as soft spherical particles, living in the multi-dimensional latent or embedding spaces, subjected to repulsive inter-particles contact forces, a random brownian force as well as a global attractive potential. Due to the similarity of this kind of dynamics to the *Langevin equation*, a stochastic differential equation (SDE) that describes the *Brownian motion* of small particles in a solvent and its generalization, we call our first algorithm *Langevin*. This algorithm allows us to approach a dense packing of the spherical identities while keeping latent space spread

## 1. Introduction

Real-world data, in particular biometric ones, can possibly contain sensitive information which raises privacy concerns from both legal and ethical specialists (Prabhakar et al., 2003; Nat, 2022). This is particularly true for a number of

---

[*]Equal contribution  [1]Idiap Research Institute, Martigny, Switzerland [2]École Polytechnique Fédérale de Lausanne (EPFL), Lausanne, Switzerland [3]Université de Lausanne (UNIL), Lausanne, Switzerland. Correspondence to: David Geissbühler <david.geissbuhler@idiap.ch>.

*Proceedings of the $42^{nd}$ International Conference on Machine Learning*, Vancouver, Canada. PMLR 267, 2025. Copyright 2025 by the author(s).

minimal and thus filling up the latent space starting by the regions that yield the most realistic images. In addition to *Langevin*, that generates the *inter-class* distributions, we also develop two similar algorithms, called *Dispersion* and *DisCo*, to generate *intra-class* variations. Figure 1 shows example of images generated with this method. We perform an exploration of the parameter space of these algorithms and generate several synthetic face datasets. We validate our approach by training FR models with this synthetic data, showing that our method can achieve a competitive performance with state of the art.

In summary, the contributions of the paper are as follows:

- We propose *physics-inspired* algorithms to generate large-scale face images datasets by sampling synthetic identities in the latent space of a generative network. We formalize our problem with *Brownian motion* of small particles in a solvent and solve it with a stochastic differential equation.

- The *Langevin* algorithm that generates a sampling of synthetic identities within a GAN's latent space optimizing *inter-class* distances, based on a loss function inspired by granular materials first used in this field.

- The *Dispersion* and *DisCo* algorithms that generate *intra-class* variations for an ensemble of synthetic identities, based on the same mechanism as well as pre-computed latent directions for the second one.

The remainder of the paper is organized as follow. We first review the existing synthetic face recognition datasets in section 2. Then, in section 3 we describe our physics-inspired methods for identity diffusion and generation of synthetic datasets. In section 4, we present experiments that validate our approach. Finally, the paper is concluded in section 5.

## 2. Related Work

Considering the legal and privacy concerns in FR models trained with large real face datasets, several works proposed new synthetic face recognition datasets (composed of different synthetic subjects with several samples per identity) to use for training face recognition models. DigiFace (Bae et al., 2023) used a computer graphic pipeline to render digital faces and introduce different variations based on face attributes (e.g., variation in facial pose, accessories, and textures). In contrast to DigiFace, other methods used generative neural networks based on GANs or diffusion models. SynFace (Qiu et al., 2021) used a modified Style-GAN2 (Deng et al., 2020) to generate synthetic images as different identities and then generated different samples by mixing identities in latent space. SFace (Boutros

et al., 2022) used CASIA-WebFace (Yi et al., 2014) to train identity-conditioned StyleGAN and then used it to generate a synthetic dataset. The similar approach was used in SFace2 (Boutros et al., 2024), but instead sampling step was performed in the latent space. In (Liang et al., 2023), a synthetic dataset consisting of 48,000 synthetic face image pairs with 555,000 human annotations is presented and used to measure bias in FR systems.

GANDiffFace (Melzi et al., 2023) used StyleGAN to generate synthetic identities and then used DreamBooth (Ruiz et al., 2023) to generate different samples for each identity. IDnet (Kolf et al., 2023) used a three-player GAN framework to generate a synthetic dataset using the Style-GAN model as a generator, and the third player is trained to generate identity-separable face images. Syn-Multi-PIE (Colbois et al., 2021) used StyleGAN to generate synthetic face images and then explored the latent space to generate different samples per identity. ExFaceGAN (Boutros et al., 2023b) used GAN-based face generator models (such as StyleGAN2 (Karras et al., 2020), StyleGAN3 (Karras et al., 2021), or GAN-Control (Shoshan et al., 2021)) and learned an identity boundary in the latent space of GAN model.

In contrast to most works based on GAN models, recently DCFace (Kim et al., 2023) and IDiff-Face (Boutros et al., 2023a) were proposed which used diffusion models to generate synthetic datasets. Unlike GAN-based face generator models like StyleGAN, the latent space of diffusion models lack style representation, and therefore DCFace (Kim et al., 2023) and IDiff-Face (Boutros et al., 2023a) trained conditional diffusion models. DCFace (Kim et al., 2023) used CASIA-WebFace (Yi et al., 2014) to train a dual condition diffusion model with style and identity conditions. IDiff-Face (Boutros et al., 2023a) trained an identity-conditioned diffusion model and used it to generate different samples of each subject. While diffusion-based methods have achieved state-of-the-art performance, diffusion models are shown to memorize individual images from their training data, and thus being less private than GAN (Carlini et al., 2023). This has led to considerable data leakage in diffusion-based synthetic face recognition datasets, such as memorization in DCFace as shown in (Shahreza & Marcel, 2024). Hence, in this paper, we focus on GAN models which are shown to be more privacy-friendly (Carlini et al., 2023).

Loss functions inspired by physical particles dynamics have already been used in the context of GANs and Diffusion Models. In (Wang et al., 2018), the authors propose a repulsive loss function based on bounded Gaussian kernel inspired by the hinge loss, showing it significantly improve training at no additional computational cost. Similarly, in (Unterthiner et al., 2017) the GAN learning problem is reformulated with a potential field inspired by electro-static forces, which allow the authors to prove that such Coulomb

GANs possess only one Nash equilibrium which help eliminate mode collapse during training. In (Franceschi et al., 2023), a framework that unifies GANs and Diffusion Models is proposed which treats GANS as interacting particle models.

## 3. Identity Diffusion and Dataset Generation

In this work, we focus on GAN generators as their latent space has a tractable dimensionality. In addition, GANs are shown to have less leakage from their training data compared to other generator models, such as diffusion models (Carlini et al., 2023). To generate a random face image on a StyleGAN type network, we can first sample a point $z$ from a Gaussian distribution in an auxiliary latent space $\mathcal{Z}$, then map it to a latent space vector $w \in \mathcal{W}$ using a fully connected mapping network and finally generate an image $i \in \mathcal{I}$ using a generator network $g(w)$. This purely Gaussian sampling, however, yields a distribution of identities that are not sufficiently dissimilar to train an FR model. The following sections present the *Langevin* algorithm that iteratively optimizes an initially random ensemble of synthetic identities towards a more suitable one where identities are distant enough. In addition to an ensemble of synthetic identities, most FR tasks also require *intra-class* variations. To this end, we propose the identity *Dispersion* algorithm which samples the latent space around the identity reference latent vector and optimize this *intra-class* ensemble in a way that samples are close in embeddings space. Finally, we present the *DisCo* algorithm which combines identity *Dispersion* with latent directions augmentation.

### 3.1. Physics Inspiration

We take here inspiration from granular mechanics and Brownian dynamics (Einstein, 1905) for our algorithms. We treat synthetic identities as soft particles which are allowed to partially overlap and where their contact interactions are modeled as a spring-like force whose magnitude is proportional to the overlap (Figure 2). This is similar to Particle Based Methods (PBMs), such as Discrete Element Method (DEM) (Cundall & Strack, 1979) which aims to simulate physical granular materials.

Each particle is modeled as a perfect sphere of diameter $d_0 = 2\,r_0$, labeled with indices $a, b, c, \cdots = 1 \ldots N$, where $\vec{x}_a = \left(x_a^i\right)$ is the position of the center of the $a$-th particle in an euclidean $D$-dimensional space $i, j, \cdots = 1 \ldots D$. The dynamics of the system are described by the Newton equation

$$m\ddot{\vec{x}}_a = \sum \vec{F}_a, \tag{1}$$

where $m$ is the mass of the particle, $\ddot{\vec{x}}_a = \frac{d^2 \vec{x}_a}{dt^2}$ its acceleration and $\sum \vec{F}_a$ the sum of the forces acting on it.

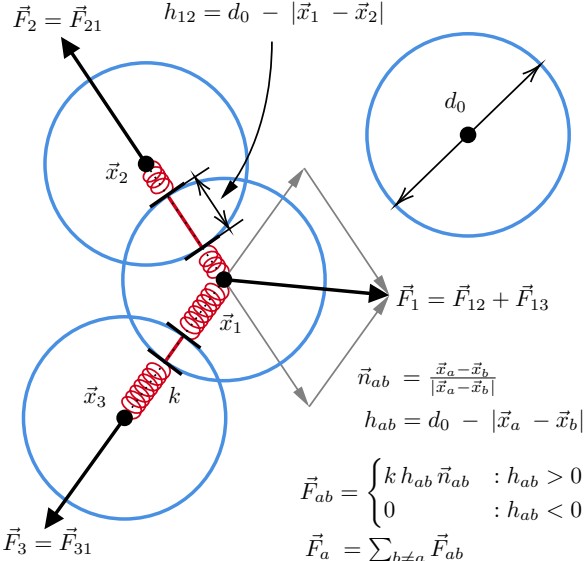

Figure 2: Simplified model of contact forces between soft spherical bodies. The bodies are characterized by their position $\vec{x}_a$ and diameter $d_0$, from which one derives the overlap $h_{ab}$ and a unit vector $\vec{n}_{ab}$ parallel to $\vec{x}_a - \vec{x}_b$. The force exerted on the body $a$ by the body $b$ is denoted $\vec{F}_{ab}$ and the total force on body $a$ by $\vec{F}_a$.

A particle $a$ is allowed to have a small overlap $h_{ab} = d_0 - |\vec{x}_a - \vec{x}_b|$ with a neighboring particle $b$. The contact forces between particles can be then derived from a quadratic spring-like potential

$$V = V(\vec{x}_1, \ldots, \vec{x}_N) = \sum_{a=1}^{N} \sum_{b=a+1}^{N} V^{\text{cont}}(\vec{x}_a, \vec{x}_b),$$

$$V^{\text{cont}}(\vec{x}_a, \vec{x}_b) = \begin{cases} \frac{k}{2} \left(d_0 - |\vec{x}_a - \vec{x}_b|\right)^2 & : |\vec{x}_a - \vec{x}_b| < d_0 \\ 0 & : |\vec{x}_a - \vec{x}_b| > d_0 \end{cases}, \tag{2}$$

where $k$ is the spring constant and $x = \{\vec{x}_1 \ldots \vec{x}_N\}$ denote the positions of the particles. The sum of the contact forces acting on the particle $a$ can be then recovered by taking minus the gradient w.r.t. the particle position $\vec{x}_a$

$$\vec{F}_a^{cont} = -\vec{\nabla}_{\vec{x}_a} V. \tag{3}$$

In this work, we use GAN models (e.g. StyleGAN) to generate synthetic identities. These generative models typically yield better quality images around a center point in latent space. To keep samples close to such a point, we introduce another spring-like force that pulls back samples toward this point

$$\vec{F}_a^{pull-back} = -k^{pull-back} \left(\vec{x}_a - \vec{x}_{avg}\right), \tag{4}$$

where $\vec{x}_{avg}$ denotes the coordinates the center point of the latent space and $k^{pull-back}$ is a spring constant. In addi-

tion to these conservative forces, we also add a dissipative viscous force to damp the system

$$\vec{F}_a^{visc} = -\mu \dot{\vec{x}}_a, \tag{5}$$

where $\mu$ is the viscous force constant and $\dot{\vec{x}}_a = \frac{d\vec{x}_a}{dt}$ the velocity of the particle.

Generally, GANs map a probability distribution in the latent space $\mathcal{Z}$ to a distribution in the target space $\mathcal{I}$

$$p(z) \longrightarrow p(i), \quad z \in \mathcal{Z}, \quad i \in \mathcal{I}, \tag{6}$$

where the latent distribution $p(z)$ is often chosen as the normal distribution $\mathcal{N}(0, \mathbb{I})$. We are interested here in a modified distribution $p'(z)$ that optimizes two simultaneous constraints: 1) The mapping of this distribution to an auxiliary identity embedding space $\mathcal{E}$ spans as much as possible of this space, 2) the variance and mean of $p'(z)$ are minimized assuring the GAN generates realistic images. We assume that there exists a dynamical process that evolves the initial distribution $p(z) = p(z, t)|_{t=0}$ toward an equilibrium solution which is the solution which optimize the constraints

$$p'(z) = p(z, t)|_{t=\infty}. \tag{7}$$

One possible way to describe such a dynamical process for the distribution $p(z, t)$ is to use generalizations of the diffusion equation, such as the Fokker-Plank equation

$$\dot{p} = -\partial_i \left( \mu^i \, p \right) + \frac{1}{2} \partial_i \partial_j \left( \sigma^{ij} \, p \right) + \dots, \tag{8}$$

where $\partial_i = \frac{\partial}{\partial z^i}$ and $\mu^i = \mu^i(z, t)$, $\sigma^{ij} = \sigma^{ij}(z, t)$ are coefficient functions that control the dynamics of the system. In practice, solving such an equation can be challenging and we choose here to re-formulate the dynamics in term of the particle degrees of freedom via the Langevin equation:

$$\mu \dot{\vec{x}}_a = \sum \vec{F}_a + \vec{\Gamma}_a(t). \tag{9}$$

This equation is a Stochastic Differential Equation (SDE) (Carmona et al., 1986; Dalang & Sanz-Solé, 2024), which can be obtained from the Newton equation (i.e., Eq. 1) by adding a time-dependent random force $\vec{\Gamma}_a(t)$. We consider here the over-damped limit of this equation where viscous forces are dominant over inertia ($\mu \gg m$) and neglect the latter ($m \to 0$). The random force is assumed not to favor any particular direction, i.e.,

$$\left\langle \vec{\Gamma}_a(t) \right\rangle = 0 \tag{10}$$

and has a Gaussian probability distribution. It is Markovian with the following temperature dependent self-correlation relation

$$\left\langle \vec{\Gamma}_a(t') \left( \vec{\Gamma}_b(t) \right)^T \right\rangle = 2\mu \, k_B T \, \mathbb{I}_{D \times D} \, \delta_{ab} \, \delta(t' - t), \tag{11}$$

where $\mu$ is the viscous coefficient, $k_B$ the Boltzmann constant and where $T$ is the temperature.

## 3.2. Identity Embeddings and Metrics

We consider a generative model $g(w)$ that maps a latent space $\mathcal{W}$ to an image space $\mathcal{I}$. In the case of the StyleGAN family of models, this network is complemented by a mapping network $f(z)$ that maps an auxiliary space $\mathcal{Z}$ where gaussian sampling is performed. In addition to this generative model, we select an off-the-shelf face recognition (FR) model $h(i)$ that extracts a face embedding vector $e \in \mathcal{E}$. To obtain an embedding from a random latent sample, one evaluates the following chain: $z_a \sim \mathcal{N}(0, \mathbb{I})$, $w_a = f(z_a)$, $i_a = g(w_a)$, $e_a = h(i_a)$. The default FR models we use in this work are all based on the ArcFace loss function (Deng et al., 2019), which has a spherical symmetry. To be consistent with this symmetry, we define an angular metric on the embedding space $\mathcal{E}$ that simply measure the angle between the vectors

$$d^{\mathcal{E}}(e_a, e_b) = \arccos \frac{e_a \cdot e_b}{|e_a| \, |e_b|}. \tag{12}$$

This in turns allows us to put a identity aware metric on the latent space

$$d_{id}^{\mathcal{W}}(w_a, w_b) = d^{\mathcal{E}}(e(w_a), e(w_b)) \tag{13}$$

where $e(w) = h(g(w))$.

## 3.3. Inter-Class Optimization

### 3.3.1. RANDOM-REJECT SAMPLING ALGORITHM

We first consider a simple identity sampling algorithm as a baseline. This algorithm works iteratively by randomly sampling a latent vector $w_{n+1}$ and then computing its face embedding $e_{n+1}$. It then compute the distances $d^{\mathcal{E}}$ between $e_{n+1}$ and the $n$ previously accepted samples $\{e_a, a = 1 \dots n\}$ and, if all these values are above an Inter-Class Threshold (ICT), the sample $w_{n+1}$ is itself accepted. If not the procedure is repeated until a sample that satisfies this criterion is found. This process is repeated until the desired number of identities $N_{id}$ is reached. This algorithm, used for instance in (Colbois et al., 2021), while perfectly suitable to find sufficiently dissimilar identities, unfortunately scales exponentially with the number of identities making it hard to apply to large datasets. We call this algorithm *Reject* in following sections.

### 3.3.2. IDENTITY SAMPLING FROM LANGEVIN DYNAMICS

To circumvent the scaling problem of the aforementioned *Reject* sampling algorithm, we present here an iterative algorithm, inspired by the physical systems presented earlier in this section. The main idea is to introduce a *repulsive force* between the embeddings $e_a = e(w_a)$ so that they naturally arrange themselves in an assembly that maximize

their inter-class distances. While any kind of repulsive force, found in the physical world or not, could in principle serve this purpose, a spring-like force that has a linear dependency on the position seems the simplest choice. Moreover, such type of forces are independent of the dimensionality of the space they act in. On the contrary, other physical forces such as gravity or electrostatics are described by power laws with exponents that depends crucially on the dimensionality of space and seem less adapted for our purpose.

Ideally, we do not want that identities that are far away interact together, only identities pairs whose distance is below a certain ICT value should lead to a repulsive force. With this criterions in mind, we observe that the potential for non-dissipative granular contact interactions in Eq. 2 achieves precisely this, as the interaction vanishes when the distance is bigger than a constant $d_0$. Based on these ideas, we define the identity granular repulsion loss

$$
\mathcal{L}^{\mathcal{E}} = \frac{k^{\mathcal{E}}}{2} \sum_{a=1}^{N_{id}} \sum_{b=a+1}^{N_{id}} \begin{cases} \left(d_0 - d_{ab}^{\mathcal{E}}\right)^2 & : d_{ab}^{\mathcal{E}} < d_0 \\ 0 & : d_{ab}^{\mathcal{E}} > d_0 \end{cases} \quad (14)
$$
$$
d_{ab}^{\mathcal{E}} = d^{\mathcal{E}}\left(e\left(w_a\right), e\left(w_b\right)\right) = d_{id}^{\mathcal{W}}\left(w_a, w_b\right).
$$

We note that this loss function is a close relative of non-linear Hinge Losses commonly used in SVM algorithms (Luo et al., 2021). Sampling a set identities that satisfy a minimal distance threshold is similar to a sphere packing problem. While solutions to this problem are known for two and three dimensions, higher dimensional optimal sphere packing lattices are usually unknown, except in some special cases (Cohn et al., 2017). Ideally, we would like our algorithm to generate the densest packing possible to get the most identities from a given generative network. In practice, as no algebraic solution can be found in such high dimensionality we assume that our dynamical optimization method yields a good enough solution. While the stochastic noise introduced in Eq. 10 is not strictly necessary to this optimization problem, we choose to keep it as it introduces an additional temperature hyper-parameter via Eq. 11 as well as a source of randomness which can be switched off if needed. An hypothetical benefit of randomness in this context is to prevent the formation of *jammed states*, which can happen in granular materials in two or three dimensions where particles are inter-locked in a way that can block further rearrangement of the granular elements (Brilliantov et al., 2004). We do not expect such jamming to appear in this high dimensionality context and indeed have not observed such a phenomenon in our limited experiments. However, we observed a minor improvement in the convergence speed. It is worth noting that jamming transitions have already been studied in the context of loss landscapes of deep neural networks (Geiger et al., 2019).

Generative models, and GANs in particular, generate good quality images when the input latent vector is in a sub-space of the full latent space. For instance, the StyleGAN family of models (Karras et al., 2019; 2020; 2021) implements the so-called *truncation trick* which consists by rescaling the latent vector by a constant factor w.r.t. an origin placed at the average latent $w_{avg}$ calculated by sampling a large number of vectors via the mapping network. In other words, the best quality images are those with latents located near $w_{avg}$. Introducing purely repulsive interactions alone would be problematic as the latent vectors would be pushed away from the domain where the network generates the best quality data. To circumvent this we introduce a latent pull-back loss function, quadratic as well, that keeps the identities from wandering too far from the $w_{avg}$ latent

$$
\mathcal{L}^{\mathcal{W}} = \frac{k^{\mathcal{W}}}{2} \sum_{a=1}^{N_{id}} |w_a - w_{avg}|^2 \quad (15)
$$

We call the resulting identity sampling algorithm *Langevin* due to its similarities with the equations describing motion a small soft particles in a thermal bath.

### 3.3.3. NUMERICAL IMPLEMENTATION

As said earlier we focus on the case where viscosity is dominant w.r.t inertia. For physical particles we thus need to solve the first order stochastic differential equation in Eq. 9, which we approximate by

$$
\vec{x}_a(t + \delta t) \approx \vec{x}_i(t) + \frac{\delta t}{\mu}\vec{F}_a + \frac{\eta_0}{\mu}d\vec{W}(t, \delta t)
$$
$$
= \vec{x}_i(t) - \frac{\delta t}{2}\frac{k}{\mu}\vec{\nabla}_{\vec{x}_a}\sum_{a,b>a}h_{ab}^2 + \frac{\eta_0}{\mu}d\vec{W}(t, \delta t)
$$
$$
(16)
$$

where we see that the viscosity constant $\mu$ scales the other constants $k$ and $\eta_0$. For the following discussion we set the viscosity to be $\mu = 1$ and use the other constants to parametrize the problem. Our latent update algorithm, inspired by the above simple numerical scheme, reads:

$$
w_a^{(t+1)} = w_a^{(t)} - \delta t\,\nabla_{w_a}\mathcal{L}^{(t)} + \eta_0\sqrt{\delta t}\,\zeta_a^{(t)}, \quad (17)
$$

where $\zeta_a^{(t)}$ is a vector of independent normal variables of variance $\sigma = 1$ and where

$$
\mathcal{L}^{(t)} = \left(\mathcal{L}^{\mathcal{E}} + \mathcal{L}^{\mathcal{W}}\right)\left(w_1^{(t)}, \ldots, w_N^{(t)}\right). \quad (18)
$$

From a numerical perspective, the gradient of $\mathcal{L}^{\mathcal{W}}$ can be easily calculated so the only challenging task is the computation of the gradient of the embedding distance metric $\nabla_a\,d_{bc}^{\mathcal{E}}$. This computation can be challenging because the embedding computation passes through two networks, the generator and the embedding extractor, and through a very high dimensional image space. One can simplify the problem by computing the jacobian $\frac{\partial e}{\partial w}$, but it is still quite computationally expensive. We find that a more efficient way

**Algorithm 1** Langevin algorithm

1: **for** $a = 1 \ldots N_{id}$ **do**
2:      $z_a \leftarrow \mathcal{N}$
3:      $w_a \leftarrow f(z_a)$
4: **end for**
5: **for** $N_{iter}$ **do**
6:      **for** $a = 1 \ldots N_{id}$ **do**
7:         $i_a \leftarrow g(w_a)$
8:         $e_a^{cst} \leftarrow h(i_a)$
9:      **end for**
10:     **for** $a = 1 \ldots N_{id}$ **do**
11:        $i_a \leftarrow g(w_a)$
12:        $e_a \leftarrow h(i_a)$
13:        $d_{ab}^{\mathcal{E}} \leftarrow d^{\mathcal{E}}(e_a, e_b^{cst}), \quad b \neq a$
14:        $f_a \leftarrow -\nabla_a \left( \mathcal{L}^{\mathcal{E}} + \mathcal{L}^{\mathcal{W}} \right)$
15:     **end for**
16:     $f^+ \leftarrow \arg\max f_a$
17:     $\delta w^- \leftarrow \arg\min |w_a - w_b|$
18:     $\delta t \leftarrow \tau \, \delta w^- / f^+$
19:     **for** $a = 1 \ldots N_{id}$ **do**
20:        $\zeta_a \leftarrow \mathcal{N}$
21:        $w_a \leftarrow w_a + \delta t f_a + \sqrt{\delta t}\zeta_a$
22:     **end for**
23: **end for**

of performing this computation is to first compute all the embeddings with a forward only pass, and then compute the gradients in a second pass. This procedure allows us to make the problem tractable on standard computing hardware, even for a large number of identities. We still need to determine a appropriate time-step for our calculations. We will show in the next section that the time-step has little impact, a wide range of values yielding acceptable performance. Values too coarse can however lead to numerical problems that can have a negative impact over the quality of the resulting dataset. Looking at Eq. 17, and neglecting the random force, we see that the maximal distance a latent can move in a single time-step, $\arg\max \left| w_a^{(t+1)} - w_a^{(t)} \right|$ is proportional to the time-step times $\arg\max \left| \nabla \mathcal{L}^{(t)} \right|$. If we impose that this distance should remain smaller than a proportion $\tau < 1$ of the minimal latent-to-latent distance, we find the following expression for the time-step

$$\delta t = \tau \frac{\arg\min |w_a - w_b|}{\arg\max \left| \nabla \mathcal{L}^{(t)} \right|}, \qquad (19)$$

which prevent latent vectors to be updated too aggressively while still giving good numerical performance when the distribution is close to an equilibrium. The final *Langevin* algorithm is depicted in Algorithm 1. An illustration of the *Langevin* algorithm is depicted in Appendix B.

## 3.4. Intra-Class Variations

### 3.4.1. IDENTITY DISPERSION

As we are interested in generating synthetic datasets that can be used to train FR models, we need variations of the synthetic identities, so called within-class variations. We devise here a second algorithm, called *Dispersion*, that generates an arbitrary number of variations from a reference latent-embedding pair $(w_a^{ref}, e_a^{ref})$ while preserving as much as possible the identity. This algorithms is similar to the *Langevin* algorithm but is intended to be used to after a suitable set of identities have been generated by the latter. Given a number $N_{id}$ of references identities, this algorithm creates $N_{var}$ variations $w_a^\alpha$, with $\alpha = 1 \ldots N_{var}$. To keep these new latent vectors separated enough, to create variability, we introduce another granular loss function, but this time in the latent space

$$\mathcal{L}_{disp}^{\mathcal{W}} = \frac{k_{disp}^{\mathcal{W}}}{2} \sum_{a=1}^{N_{id}} \sum_{\alpha=1}^{N_{var}} \sum_{\beta=\alpha+1}^{N_{var}}$$
$$\begin{cases} \left( d_0^{\mathcal{W}} - |w_a^\alpha - w_a^\beta| \right)^2 & : |w_a^\alpha - w_a^\beta| < d_0^w \\ 0 & : |w_a^\alpha - w_a^\beta| > d_0^w \end{cases} \qquad (20)$$

For the present work, we do not compute contact forces between different identities for identity variations, even if this could potentially improve performance. This is done mainly for simplicity reasons as such extension would require parallelization with custom inter-GPU communications. To keep the embeddings of the identities variations close to the reference embedding, we introduce a further spring-like quadratic loss function

$$\mathcal{L}_{disp}^{\mathcal{E}} = \frac{k_{disp}^{\mathcal{E}}}{2} \sum_{a=1}^{N_{id}} \sum_{\alpha=1}^{N_{var}} d^{\mathcal{E}} \left( e\left( w_a^\alpha \right) , e_a^{ref} \right)^2, \qquad (21)$$

which is computed similarly to the granular loss of the *Langevin* algorithm. Finally, we also add the latent pull-back loss in Eq. 15, the total *Dispersion* loss reads

$$\mathcal{L}_{disp} = \mathcal{L}_{disp}^{\mathcal{W}} + \mathcal{L}_{disp}^{\mathcal{E}} + \mathcal{L}^{\mathcal{W}}. \qquad (22)$$

The numerical implementation of this algorithm is very similar to the *Langevin* one, the latent update equation simply reads

$$w_a^{\alpha \, (t+1)} = w_a^{\alpha \, (t)} - \delta t \, \nabla_{w_a^\alpha} \mathcal{L}_{disp}^{(t)} + \eta_0 \sqrt{\delta t} \, \zeta_a^{\alpha \, (t)}. \qquad (23)$$

In this article we keep a fixed time-step for *Dispersion*, for simplicity reasons, and parallelize over identities. We initialize the new latents $w_a^\alpha$ with the value of the reference one $w_a^{ref}$ and add some random gaussian noise to break the symmetry of the ensemble

$$w_a^{\alpha \, (0)} = w_a^{ref} + \xi_0 \, \xi_a^\alpha, \qquad (24)$$

where $\xi_a^\alpha$ is a vector of independent normal variables of variance $\sigma = 1$ and $\xi_0$ is a fixed scaling parameter. An illustration of the *Dispersion* algorithm is depicted in Appendix B.

### 3.4.2. LATENT EDITING: COVARIATES

While the *Dispersion* algorithm creates realistic variations of a given identity, it gives little control over the type of variation created. Moreover, having an alternative method to create variations is desirable to give a comparison point. For this reason we use the latent editing method introduced in (Colbois et al., 2021). This latent editing technique assumes that the variation of some attribute, such as left-right pose, is essentially a translation in latent space and that this translation is the same for every identity.

Given this assumption and following (Colbois et al., 2021), we project in the latent space the samples of the CMU Multi-PIE face database. This dataset provides a reasonable number of genuine identities, each captured with different poses, illuminations and expressions. When this is done, we use a linear SVM model to fit the latent directions corresponding to the different attributes variations present in the database. In particular, we extract 7 vectors $w_I^{cov}$ corresponding to left-right poses, left-right illuminations and 5 facial expressions: smile, surprise, squint, disgust and scream. The algorithm introduced in (Colbois et al., 2021) based on these latent direction is called *Covariates* in this work and yields a total of 17 variations: 6 different poses, 6 different illuminations and one for each of the 5 expressions.

### 3.4.3. COMBINING DISPERSION AND COVARIATES

We also propose to combine the *Dispersion* and *Covariates* methods and name the resulting algorithm *DisCo*. The essential difference with the original *Dispersion* algorithm is in the initialization procedure. In addition to the initial symmetry breaking gaussian noise $\xi_a^\alpha$, the *DisCo* algorithm also adds a linear combination of the 7 *Covariates* vectors $w_I^{cov}$ to the reference latents $w_a^{ref}$

$$w_a^{\alpha\,(0)} = w_a^{ref} + \xi_0\,\xi_a^\alpha + \sum_{I=1}^{7} \lambda_{aI}^\alpha\,w_I^{cov}, \qquad (25)$$

where weights $\lambda_{aI}^\alpha \in [-\lambda_0, \lambda_0]$ are randomly and uniformly sampled in a seven-dimensional hypercube. This additional step forces more intra-class variability at the initial step of the *Dispersion* algorithm and yields, according to our experiments, better performing datasets. This positive effect might be explained by the number of identities variations compared to the dimensionality of the intra-class latent subspace. In the case where the former is small compared to the latter, this extra initialization step helps the latent granular contact loss to spread the latent vectors across the intra-class subspace, yielding a more diverse final dataset.

## 4. Experiments

### 4.1. Experimental Setup

**Training and Benchmarking**  We use the synthetic dataset and train a face recognition model with the IResNet50 backbone using AdaFace loss function (Kim et al., 2022). We train each model for 30 epochs using the Stochastic Gradient Descent (SGD) optimizer with the initial learning rate 0.1 and weight decay $5 \times 10^{-4}$. Then, we evaluate the performance of the trained models on different benchmarking datasets, including Labeled Faces in the Wild (LFW) (Huang et al., 2008), Cross-age LFW (CA-LFW) (Zheng et al., 2017), CrossPose LFW (CP-LFW) (Zheng & Deng, 2018), Celebrities in Frontal-Profile in the Wild (CFP-FP) (Sengupta et al., 2016), and AgeDB-30 (Moschoglou et al., 2017) datasets. To maintain consistency with prior works, the results reported for LFW, CA-LFW, CP-LFW, CFP-FP, and AgeDB datasets are accurately calculated using 10-fold cross-validation, where the comparison threshold is set at the Equal Error Rate (ERR) on one fold and the accuracy is measured on the remaining folds.

**Different Hyperparameters**  Our new method introduces a number of hyperparameters that influence the quality and usefulness of the final synthetic datasets. Table 5 in Appendix C shows the list of hyperparameters of the *Langevin*, *Dispersion* and *DisCo* algorithms as well as their default values. We also provide ablation study with different values and evaluate the resulting datasets.

**Reference FR Backbone**  To compute embedding distances and optimize the identity distribution with respect to the latter, we choose an off-the-shelf reference FR model. This model is built on an IResNet50 backbone, with an ArcFace loss (Deng et al., 2019) and trained on MS-Celeb-1M dataset (Guo et al., 2016).

### 4.2. Comparison with Previous Synthetic Datasets

To compare the performance of our generated synthetic datasets with previous synthetic face recognition datasets in the literature, we train a face recognition model with same backbone and using same learning loss as explained in section 4.1. Table 1 reports the performance of face recognition model trained with different datasets. The results of benchmarking in Table 1 show that our method achieves superior performance compared to GAN-based methods. Compared to diffusion-based datasets, our method outperforms IDiff-Face on all benchmarking datasets and achieves a competitive performance with DCFace. However, we should note that diffusion models are shown to be prone to leaking information from their training samples (Carlini et al., 2023; Vyas et al., 2023; Somepalli et al., 2023a;b; Li et al., 2024; Shahreza & Marcel, 2024), which limits their

Table 1: Comparison of the existing synthetic face datasets present in the literature with the best performing datasets created in this work. In addition to the number of identities and number of images, we present the recognition accuracy obtained by training an FR model on each dataset and evaluating on standard face recognition benchmarking datasets. The best value in each category is in **bold** text and the best results achieved by training from synthetic datasets amongst all categories of synthetic datasets are underlined.

| Dataset Type | Dataset name | Generator | $N_{id}$ | $N_{samples}$ | LFW | CPLFW | CALFW | CFP | AgeDB |
|---|---|---|---|---|---|---|---|---|---|
| Real images | MS-Celeb-1M (Guo et al., 2016) | N/A | 85'000 | 5'800'000 | **99.82** | 92.83 | **96.07** | 96.10 | **97.82** |
| | WebFace-4M (Zhu et al., 2021) | N/A | 206'000 | 4'000'000 | 99.78 | **94.17** | 95.98 | **97.14** | 97.78 |
| | CASIA-WebFace (Yi et al., 2014) | N/A | 10'572 | 490'623 | 99.42 | 90.02 | 93.43 | 94.97 | 94.32 |
| Computer Graphics | DigiFace-1M (Bae et al., 2023) | Rendered mesh | 109'999 | 1'219'995 | 90.68 | 72.55 | 73.75 | 79.43 | 68.43 |
| Diffusion-based | DCFace-0.5M (Kim et al., 2023) | custom trained | 10'000 | 500'000 | 98.35 | 83.12 | 91.70 | 88.43 | 89.50 |
| | DCFace-1.2M (Kim et al., 2023) | custom trained | 60'000 | 1'200'000 | 98.90 | 84.97 | 92.80 | 89.04 | 91.52 |
| | IDiff-Face (Uniform) (Boutros et al., 2023a) | custom trained | 10'049 | 502'450 | 98.18 | 80.87 | 90.82 | 82.96 | 85.50 |
| | IDiff-Face (Two-Stage) (Boutros et al., 2023a) | custom trained | 10'050 | 502'500 | 98.00 | 77.77 | 88.55 | 82.57 | 82.35 |
| GAN-based | Synface (Qiu et al., 2021) | StyleGAN2 [b] | 10'000 | 999'994 | 86.57 | 65.10 | 70.08 | 66.79 | 59.13 |
| | SFace (Boutros et al., 2022) | StyleGAN2 [♯] | 10'572 | 1'885'877 | 93.65 | 74.90 | **80.97** | 75.36 | 70.32 |
| | SFace2 (Boutros et al., 2024) | StyleGAN2 [♯] | 10'572 | 1'048'255 | 94.03 | 73.2 | 80.33 | 74.87 | **72.98** |
| | Syn-Multi-PIE [†] (Colbois et al., 2021) | StyleGAN2 | 10'000 | 180'000 | 78.72 | 60.22 | 61.83 | 60.84 | 54.05 |
| | GANDiffFace (Melzi et al., 2023) | StyleGAN3 | 10'080 | 543'893 | **94.35** | **76.15** | 79.90 | **78.99** | 69.82 |
| | IDnet (Kolf et al., 2023) | StyleGAN2 [♯] | 10'577 | 1'057'200 | 84.48 | 68.12 | 71.42 | 68.93 | 62.63 |
| | ExFaceGAN (Boutros et al., 2023b) | GAN-Control | 10'000 | 599'944 | 85.98 | 66.97 | 70.00 | 66.96 | 57.37 |
| | *Langevin-Dispersion* [**ours**] | StyleGAN2 | 10'000 | 650'000 | 94.38 | 65.75 | 86.03 | 65.51 | 77.30 |
| | *Langevin-DisCo* [**ours**] | StyleGAN2 | 10'000 | 650'000 | 97.07 | 76.73 | 89.05 | 79.56 | 83.38 |
| | *Langevin-DisCo* [**ours**] | StyleGAN2 | 30'000 | 1'950'000 | 98.97 | 81.52 | 93.95 | 83.77 | 93.32 |

[†] Dataset was re-generated from the original source code    [♯] Identity conditioned    [b] Disentangled representation (Deng et al., 2020)

Table 2: Influence of the number of *Langevin* iterations $N_{iter}$ on FR accuracy. For all datasets, variations are created via the *Dispersion* algorithm with default parameters. The first row consider pure random sampling for benchmarking. The next rows show different combination of $N_{iter}$ and $d_0^{\mathcal{E}}$.

| $N_{iter}$ | $d_0^{\mathcal{E}}$ | LFW | CPLFW | CALFW | CFP | AgeDB | Average |
|---|---|---|---|---|---|---|---|
| 0 | - | **90.78** | **62.55** | **78.5** | **64.96** | **70.12** | **73.38** |
| 1 | 1.54 | 90.73 | 64.15 | 78.5 | 65.24 | 70.67 | 73.86 |
| 10 | 1.54 | 92.23 | 65.05 | 81.42 | 65.5 | 72.73 | 75.39 |
| 20 | 1.54 | 92.73 | 66.37 | 82.42 | 65.09 | 74.22 | 76.17 |
| 50 | 1.54 | **93.93** | **66.83** | 84.52 | **68.67** | 75.52 | **77.89** |
| 50 | 1.4 | 94.45 | **65.58** | 86.03 | **66.53** | 77.17 | 77.95 |
| 100 | 1.4 | **94.72** | 64.58 | **86.17** | 65.36 | **79.25** | **78.02** |

Table 3: Influence of *Langevin* time-step parameters on final FR accuracy. The parameter $\delta t$ is the fixed time-step value while $\tau$ is the automatic time-step parameter.

| $\delta t$ | $\tau$ | $N_{iter}$ | LFW | CPLFW | CALFW | CFP | AgeDB | Average |
|---|---|---|---|---|---|---|---|---|
| 0.1 | - | 200 | **84.3** | 56.63 | **72.33** | 57.49 | **65.63** | 67.28 |
| 0.3 | - | 66 | 84.12 | 57.27 | 71.15 | **58.63** | 64.98 | 67.23 |
| 0.6 | - | 33 | 83.05 | 57.65 | 69.75 | 57.91 | 63.7 | 66.41 |
| 1.0 | - | 20 | 82.9 | 57.7 | 69.9 | 57.4 | 61.23 | 65.83 |
| - | 0.3 | 50 | 84.12 | **59.03** | 72.18 | 57.8 | 63.83 | 67.39 |
| - | 1.0 | 50 | 82.83 | 57.73 | 70.67 | 58.41 | 63.57 | 66.64 |

application in tasks with sensitive data. In fact, the main motivation for generating synthetic datasets is to resolve the privacy concerns in large-scale real face recognition datasets. However, if the generated synthetic dataset has a leakage of information from a real dataset, it will have similar privacy issues. This issue, as well as possible mitigation within our framework, is further discussed in Appendix H. The results in Table 1 also show that there is still a gap between training with synthetic and real face recognition datasets.

### 4.3. Performance of the Langevin Algorithms

We are interested in the influence of the number of *Langevin* iterations on the quality of the resulting datasets and, importantly, if the latter improve FR accuracy when compared to

random sampling. Table 2 shows the accuracy of FR models trained on synthetic datasets generated with such a varying number of iterations and for two values of $d_0^{\mathcal{E}}$. The datasets in this table are composed of $N = 10k$ synthetic identities, the first one being the random sampling benchmark. For all datasets, the 64 variations are created with the *Dispersion* algorithm with the default parameters reported in Appendix E. It is clear that the *Langevin* algorithm yields a significant performance improvement, even after a small number of iterations, over pure random sampling. It is interesting that more iterations tend to yield better performance and that this improvement effect continues even while the average embedding distance has reached a plateau.

While the *Langevin* algorithm seems to yields a significant accuracy advantage for synthetic data trained FR models, it is quite computationally expensive. A straightforward parameter to optimize is the time-step value, which control

Table 4: Influence of the stochastic force on FR accuracy.

| $\eta_0$ | $N_{iter}$ | LFW | CPLFW | CALFW | CFP | AgeDB | Average |
|---|---|---|---|---|---|---|---|
| 0.003 | 100 | **94.90** | 65.4 | **86.63** | **65.9** | **79.35** | **78.44** |
| 0.01 | 100 | 94.72 | 64.58 | 86.17 | 65.36 | 79.25 | 78.02 |
| 0.03 | 100 | 94.53 | **65.67** | 85.72 | 64.99 | 79.22 | 78.03 |

the precision of the numerical integration. Table 3 shows the effect of the time-step on the final FR accuracy and shows a comparison of fixed time-step against the automatically calculated values. A small advantage is observed for smaller fixed values, at least until $\delta t = 0.3$, and calculated values with $\tau = 0.3$ reach similar performance with slightly less iterations. For this reason, a varying time-step with $\tau = 0.3$ is used in the rest of the article.

Finally, we briefly study the impact of the amplitude of the random stochastic force on the resulting FR performance. The addition of this term was motivated in section 3 by the idea that it could mitigate potential jamming and help sampling the latent space efficiently. Table 4 shows a very limited survey for this purpose, with three different values of $\eta_0$. This data shows that this parameter, at least in this very limited range, has a limited impact on the results. We report further experiments in Appendices D-K, including complexity evaluation, dynamical evolution of *Langevin* ensembles, strict *Inter-Class Threshold* constraints, repulsion distances threshold, identity leakage from training set, synthetic dataset scaling, bias evaluation, and closing the gap with real data.

## 5. Conclusion

We proposed algorithms to generate synthetic face images datasets by sampling synthetic identities in the latent space of a generative network. We introduced three complementary algorithms, *Langevin*, *Dispersion* and *DisCo*, aimed at generating large synthetic datasets to mitigate ethical issues arising from the usage of real datasets. These algorithms take inspiration from physical processes such as *Brownian motion* and *granular mechanics* to generate ensembles of samples in the latent space of generative models. To our knowledge, our use of loss functions inspired by granular mechanics is novel in this context and open promising avenues, possibly beyond face recognition applications. Importantly, these algorithms introduce a set of free parameters that control the distribution of samples in the latent space and which can be optimized to yield high quality synthetic datasets tailored for specific usages. To validate the soundness of our approach and evaluate its effectiveness at generating useful synthetic face data, we trained FR models on the synthetic datasets we generated and evaluated the resulting models on seven standard face biometric benchmarks. We also outline some future directions in Appendix L.

## Source Code and Data Availability

To adhere to the standards of reproducible research, the code used for our experiments is publicly available. In addition, we also provide access to our synthetic datasets. Project page: https://www.idiap.ch/paper/synthetics-disco

## Acknowledgment

This research is based upon work supported by CITeR (Center for Identification Technology Research) project LE-GAL2 (CITeR-21F-02i-M). This work was also funded by the Hasler foundation through the Responsible Face Recognition (SAFER) project as well as the H2020 TReSPAsS-ETN Marie Skłodowska-Curie early training network (grant agreement 860813).

## Impact Statement

Large-scale face recognition datasets are collected by crawling the web (without individuals' consent), raising ethical and privacy concerns. With the advancement of generative models, recently synthetic data has emerged as a promising solution. In this paper, we proposed a new approach to generate synthetic face recognition datasets to address privacy and ethical concerns in web-crawled real face recognition datasets. Our experiments demonstrate effectiveness of our generated datasets for training face recognition models.

We would like highlight that there are different components in our data generation pipeline, which still require real face datasets, including face generator model (StyleGAN) and pretrained face recognition model. We studied the leakage of identity in the generated datasets in Appendix H. In addition, in Appendix H, we introduced repulsive forces to push synthetic samples away from training samples.

We should also note that there might be a potential lack of diversity in different demography groups, stemming from implicit biases of the datasets used for training in our pipeline (such as the pretrained face recognition model, StyleGAN, etc.). We evaluate bias in the performance of face recognition model trained with our dataset in Appendix J. It is also noteworthy that the project on which the work has been conducted has passed an Institutional Ethical Review Board (IRB).

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

## A. Sample Images

In this appendix, we show sample images generated with our algorithms. Figure 3 shows synthetic identities obtained with the *Langevin* algorithm. Figure 4, Figure 5 and Figure 6 show *intra-class* variations of synthetic identities generated with the *Dispersion*, *Covariates* and *DisCo* algorithms, respectively.

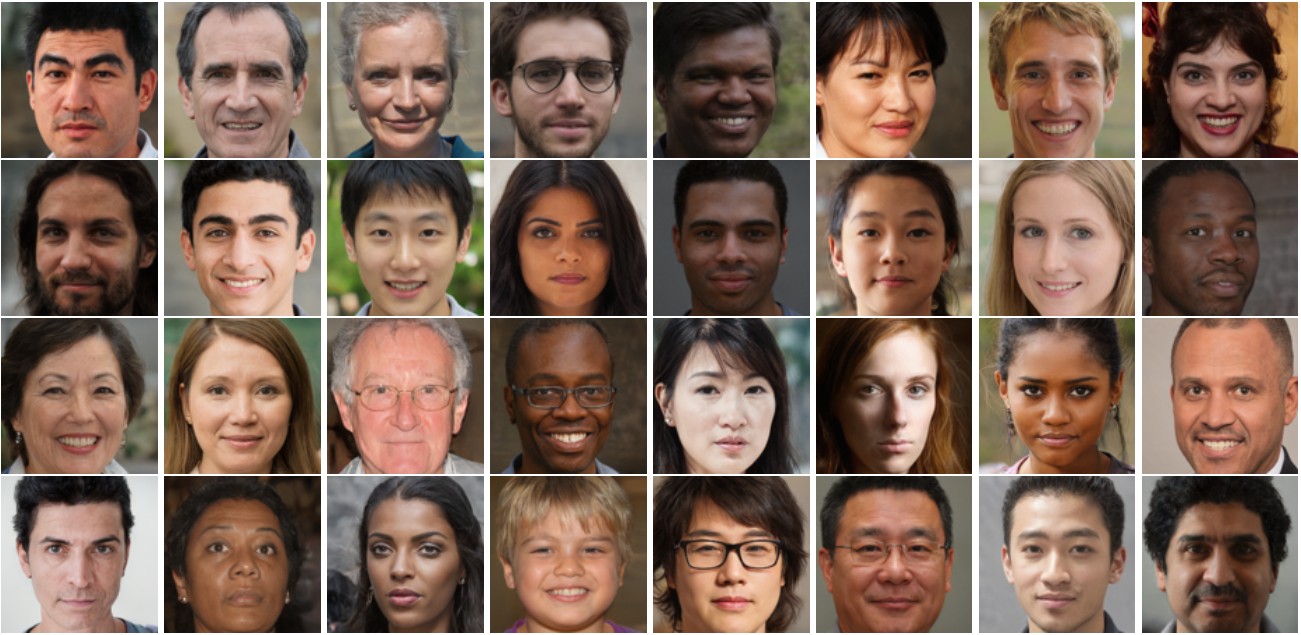

Figure 3: Sample images illustrating the *inter-class* variation produced by the *Langevin* algorithm.

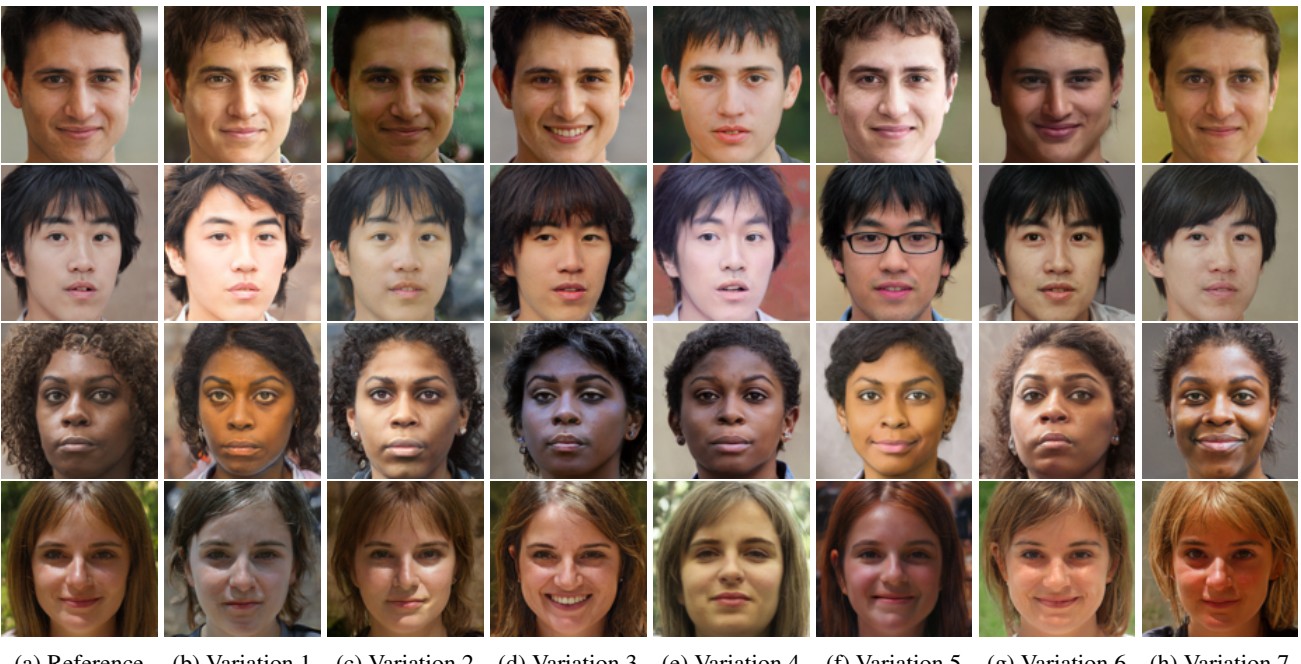

(a) Reference    (b) Variation 1    (c) Variation 2    (d) Variation 3    (e) Variation 4    (f) Variation 5    (g) Variation 6    (h) Variation 7

Figure 4: Sample images illustrating the *intra-class* variation produced by the *Dispersion* algorithm.

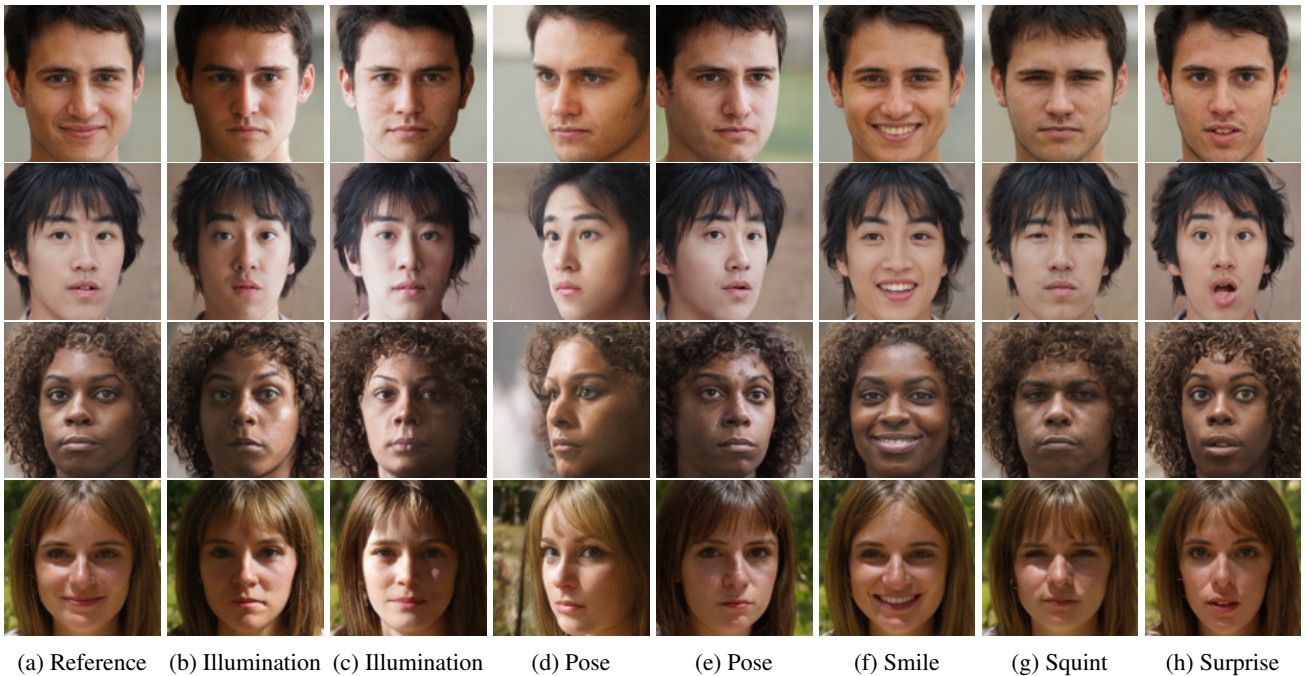

(a) Reference  (b) Illumination  (c) Illumination  (d) Pose  (e) Pose  (f) Smile  (g) Squint  (h) Surprise

Figure 5: Sample images illustrating the *intra-class* variation produced by the *Covariates* algorithm.

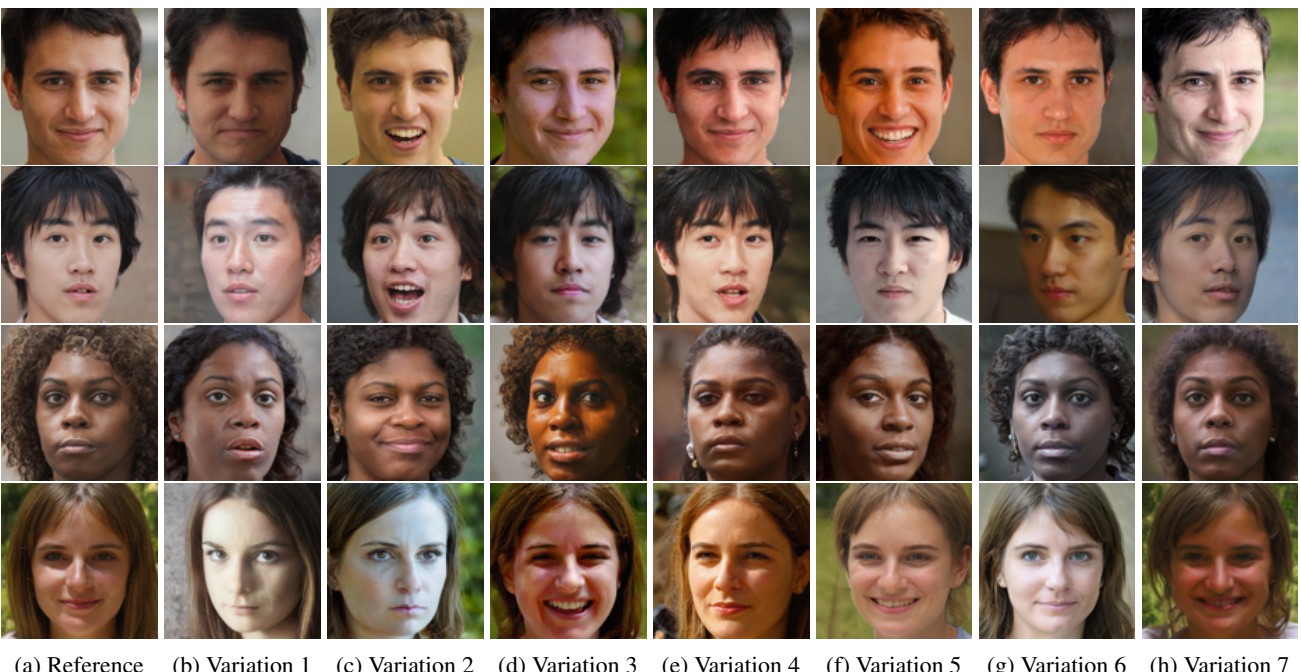

(a) Reference  (b) Variation 1  (c) Variation 2  (d) Variation 3  (e) Variation 4  (f) Variation 5  (g) Variation 6  (h) Variation 7

Figure 6: Sample images illustrating the *intra-class* variation produced by the *DisCo* algorithm.

## B. Illustrations of Langevin and Dispersion Algorithms

To generate identities with suitable *inter-class* distance between samples, we use the *Langevin* algorithm illustrated in Figure 7. This algorithm iteratively optimizes an ensemble of latent vectors, one per identity, living in the latent space of a GAN generator, using a repulsive loss function. The distance between samples is computed with an off-the-shelf FR model by

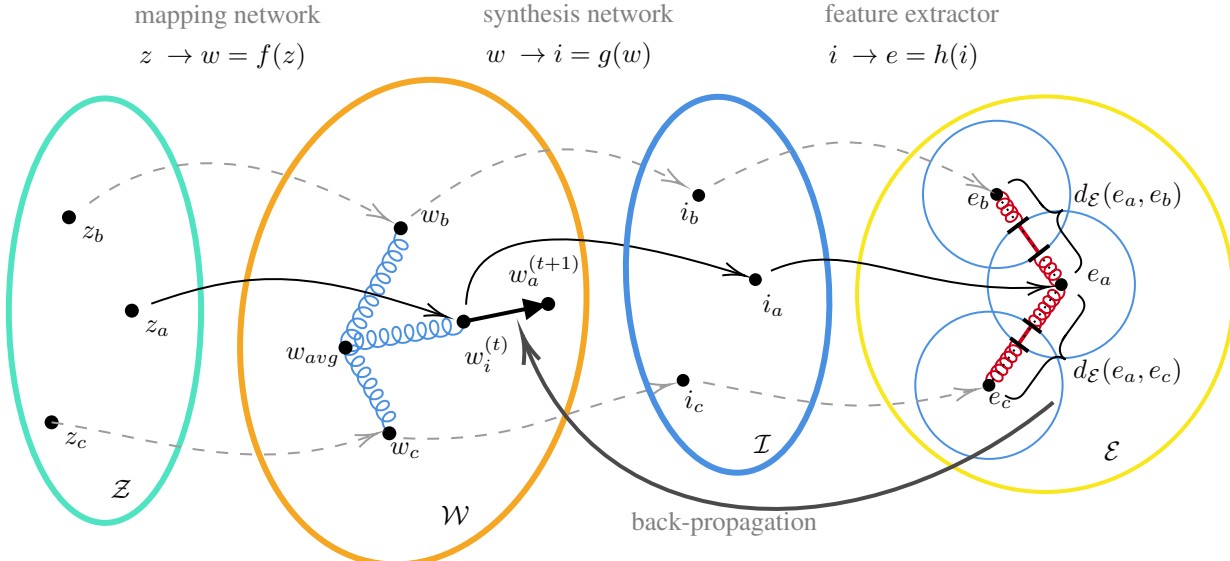

Figure 7: Mappings between the different spaces and details of the *Langevin* algorithm. Firstly, a random vector $z_a$ is sampled from a normal distribution in $\mathcal{Z}$, for each identity class $a = 1 \ldots N_{id}$. It is then mapped to the initial latent $w_a^{(0)} \in \mathcal{W}$ via the *mapping network* $f$. A face image is generated from this latent using the generator *synthesis network* $i_a = g(w_a)$ and, after face alignment, the face embedding is computed with the reference *feature extractor* $e_a = h(i_a)$. Two quadratic loss functions are introduced, one on the embedding space $\mathcal{E}$, depending on the embedding distance $d^{\mathcal{E}}$, and one on the latent space $\mathcal{W}$ distances $d^{\mathcal{W}}$. Their derivatives are evaluated using back-propagation and the latents are updated based on these gradients $w_a^{(t)} \to w_a^{(t+1)}$. The procedure is repeated for a desired number of iterations $N_{iter}$.

computing the angle between the embedding vectors. Similarly, Figure 8 depicts the *Dispersion* algorithm we use to generate *intra-class* variations of a given synthetic identity. This algorithm also iteratively optimizes the samples distribution, with the main difference that the repulsive loss function is defined in the latent space. An attractive loss function in embedding space is added to keep the identities variations as close as possible to the reference synthetic identity. The combination of these two loss functions generates *intra-class* variations that are as close as possible of the reference embedding while pushing them as far as possible in the latent space. For both *Langevin* and *Dispersion* algorithms, we add an attractive loss function in latent space that keeps the latent vectors close to the average latent, where images have the best quality.

## C. Default Values of Hyperparameters used in Experiments

Our new method introduces a number of hyperparameters that influence the quality and usefulness of the final synthetic datasets. While this gives new opportunities to tune a synthetic dataset towards a particular goal, it also adds a new layer of complexity that requires a good understanding of the influence of each of these numbers on the final results. Table 5 of this appendix shows the list of hyperparameters of the *Langevin*, *Dispersion* and *DisCo* algorithms as well as their default values. These default values give a baseline with good numerical performance and are used in the following experiments unless explicitly specified. To better understand the impact of these hyperparameters, we run a non-exhaustive survey with different values and evaluate the resulting

## D. Complexity and Required Computational Resources

The computation for the algorithms can be separated in three main phases: 1) a forward pass to generate embeddings 2) a second forward pass with backward pass to compute interactions and 3) an update step where latent vectors are updated. The run time for the *Langevin* and *DisCo* algorithms is reported in Table 6 and Table 7, respectively, for a system equipped with a single NVIDIA RTX 3090 GPU. The first pass has linear complexity with respect to the number of identities $\mathcal{O}\left(N_{id}\right)$ and can easily be parallelized. We use a batch size of 64 for this phase, which is dictated by memory constraints. The second phase has quadratic complexity $\mathcal{O}\left(N_{id}^2\right)$ as pairwise interactions are computed for each pair of samples. Similarly,

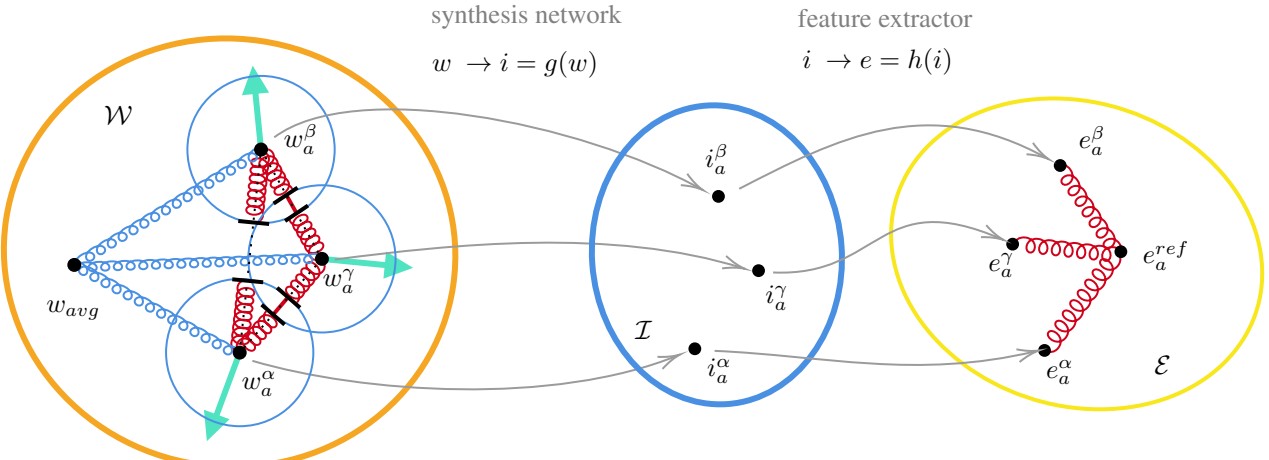

synthesis network

$w \rightarrow i = g(w)$

feature extractor

$i \rightarrow e = h(i)$

Figure 8: The *Dispersion* algorithm is very similar to *Langevin*, with slightly different loss functions, and is intended to generate $N_{var}$ *intra-class* variations per identity class. In a first step, for each variation $\alpha = 1 \ldots N_{var}$, a latent vector $w_a^{\alpha\,(0)}$ is initialized from its reference value $w_a^{ref}$ plus some noise. Three loss function are then computed. The first one acts on the embedding space and pulls the embeddings of the variations towards the reference embedding $e_a^{ref}$. The second loss function act on the latent space and pulls the latent vectors towards the average latent $w_{avg}$. The last one is a granular loss function that exert a repulsive force between latent vectors that are closer than a certain threshold. Latent vectors are updated according to the gradients of these losses. The procedure is repeated for each class $a = 1 \ldots N_{id}$ and for a desired number of iterations $N_{iter}^{disp}$.

the complexity of the second phase of the *DisCo* algorithm is linear for the number of identities $\mathcal{O}\left(N_{id}\right)$, as there are no interactions between identities, and quadratic for the number of variations $\mathcal{O}\left(N_{var}^2\right)$. In DEM and other particle-based methods, this can be reduced to linear complexity by computing a near-neighbors list at regular intervals. However, for simplicity in the implementations, we did not follow this approach. Finally, the last pass that updates the latent vectors has linear complexity $\mathcal{O}\left(N_{id}\right)$ and is very fast compared to the previous phases.

## E. Dynamical Evolution of *Langevin* Ensembles

The *Langevin* algorithm presented in the previous section is designed to maximize, iteratively and stochastically, the pairwise embedding distances of an ensemble of synthetic identities defined on the latent space of a given generative model. This is achieved using the loss function, Eq. 14 of the paper, that yields a repulsive force between two samples whose embeddings are closer than a threshold value $d_0^{\mathcal{E}}$. At the same time, a second loss function, Eq. 15 of the paper, pulls the samples towards the average latent vector $w_{avg}$ around which the best quality samples are located.

By design, the *Langevin* algorithm tries to increase the samples embedding pairwise distances $d^{\mathcal{E}}\left(e_a, e_b\right)$, up to a given threshold $d_0^{\mathcal{E}}$, while simultaneously pulling the samples towards $w_{avg}$, and therefore minimizing pairwise latent distances $|w_a - w_b|$. Figure 9a and Figure 9b show the evolution of average pairwise embedding and latent distances, respectively, for $N_{id} = 10k$ identities, up to $N_{iter} = 50$ iterations and for five different values of $d_0^{\mathcal{E}}$. Figure 9a shows that all five ensembles start with the same average pairwise embedding distance $\langle d^{\mathcal{E}} \rangle \simeq 1.47$, this quantity increase very quickly to reach a plateau after approximately 10 time-steps. As expected, bigger values of $d_0^{\mathcal{E}}$ lead to higher plateau, with the exception of $d_0^{\mathcal{E}} = 1.6$ where some sort of saturation phenomenon seems to occur. While average the embedding distance stays almost constant after this swift onset, the average pairwise latent distance $\langle |w_a - w_b| \rangle$ continue to decrease at a much slower pace, as seen in Figure 9b, indicating that the ensemble slowly clusters itself around $w_{avg}$.

These dynamics are driven by the balance between the repulsive embedding contact force, whose evolution is shown in Figure 9c, and the attractive latent pull-back force, whose evolution is shown in Figure 9d. We see that, when the samples are randomly distributed, their overlap in embedding space is quite significant leading to very high contact forces in the first iterations. The samples re-arrange themselves to minimize contact interactions quite rapidly and, after a certain number of iterations, the contact interaction reaches a plateau. After this plateau is reached, samples continue to be pulled towards

Table 5: Hyperparameters for the *Langevin*, *Dispersion* and *DisCo* algorithms and their default values.

| | Default | Description |
|---|---|---|
| $\mu$ | 1.0 | *Langevin* and *Dispersion* bulk viscosity |
| $k^{\mathcal{E}}$ | 1.0 | *Langevin* embedding contact force coefficient |
| $d_0^{\mathcal{E}}$ | 1.4 | *Langevin* embedding distance threshold |
| $k^{\mathcal{W}}$ | 0.1 | *Langevin* latent pull-back coefficient |
| $\eta_0$ | 0.01 | *Langevin* random force magnitude |
| $\tau$ | 0.3 | *Langevin* variable time-step coefficient |
| $N_{iter}$ | 100 | *Langevin* number of iterations |
| $k_{disp}^{\mathcal{W}}$ | 1.0 | *Dispersion* latent contact force coefficient |
| $d_0^{\mathcal{W}}$ | 12.0 | *Dispersion* latent distance threshold |
| $k_{disp}^{\mathcal{E}}$ | 1.0 | *Dispersion* identity pull-back coefficient |
| $\tilde{k}^{\mathcal{W}}$ | 1.0 | *Dispersion* latent pull-back coefficient |
| $\tilde{\eta}_0$ | 0.01 | *Dispersion* random force magnitude |
| $\tilde{\delta}t$ | 0.05 | *Dispersion* fixed time-step |
| $\tilde{N}_{iter}$ | 20 | *Dispersion* number of iterations |
| $\xi_0$ | 0.2 | *Dispersion* initial symmetry breaking noise |
| $\lambda_0$ | 1.0 | *DisCo* covariates sampling scale |

Table 6: Runtime for *Langevin* identities generation with $N_{iter} = 100$ on a system with a single NVIDIA RTX 3090 GPU.

| $N_{id}$ | Runtime |
|---|---|
| $1k$ | $2.2h$ |
| $2k$ | $3.5h$ |
| $4k$ | $5.4h$ |
| $6k$ | $10.2h$ |
| $8k$ | $12.8h$ |
| $10k$ | $18.2h$ |
| $20k$ | $31.5h$ |
| $30k$ | $49.2h$ |

Table 7: Runtime for *DisCo* identities variations generation with $N_{iter} = 30$ and $N_{id} = 30k$ on a system with a single NVIDIA RTX 3090 GPU.

| $N_{var}$ | Runtime |
|---|---|
| 16 | $50.0h$ |
| 32 | $95.8h$ |
| 64 | $183.7h$ |

$w_{avg}$ by the latent pull-back force, which slowly decays as seen Figure 9d. Figure 9e and Figure 9f show this decay for a varying number of identities, by plotting $\langle |w_a - w_b| \rangle$ in function of $N_{id}$ and $N_{iter}$, respectively.

## F. Strict *Inter-Class Threshold* Constraints

Another interesting metric on the performance of the algorithm is the proportion of pairwise distances that are above a certain *inter-class threshold* $d_{ict}^{\mathcal{E}}$. This can be computed by counting each occurrence where this condition is met, for each possible pair of identities $(a, b)$

$$\rho_{ict} = \frac{\left| \left\{ d^{\mathcal{E}} (e_a, e_b) < d_{ict}^{\mathcal{E}}, \, \forall (a,b), \, a > b \right\} \right|}{N_{pairs}}, \tag{26}$$

where we set $a > b$ to avoid double counting and where $N_{pairs} = \frac{N_{id}(N_{id}-1)}{2}$ is the number of possible pairwise interactions. If we set this threshold equal to the *Langevin* repulsion distance threshold, $d_{ict}^{\mathcal{E}} = d_0^{\mathcal{E}}$, we can evaluate the ratio of the number of contacts over the number of pairs

$$\rho_0 = \frac{\left| \left\{ d^{\mathcal{E}} (e_a, e_b) < d_0^{\mathcal{E}}, \, \forall (a,b), \, a > b \right\} \right|}{N_{pairs}} = \frac{N_{contacts}}{N_{pairs}}. \tag{27}$$

Figure 9g shows the dynamical evolution of this quantity after a certain number of iterations $N_{iter}$, for different values of $d_0^{\mathcal{E}}$. We see that for very high values of $d_0^{\mathcal{E}}$, this value is close to one so almost every identity is in contact with every other one. For other values we can compute the average number of contacts per identity

$$\frac{2 \, N_{contacts}}{N_{id}} = \frac{2 \, \rho_0 \, N_{pairs}}{N_{id}} = 2\rho_0 \, (N_{id} - 1), \tag{28}$$

where we have added a factor of two to account for each identity in the pair in contact. We see in Figure 9g that, for $d_0^{\mathcal{E}} = 1.4$ and $N_{id} = 10k$, this ratio stabilizes around $\rho = 0.03$ meaning that each identity is on average in contact with approximately 300 other classes. More generally, the ratio $\rho_{ict}$ is the FMR of the reference FR backbone evaluated on the synthetic dataset and the *Langevin* algorithm tends to iteratively decrease this value until it reaches a plateau.

It would be interesting to know the maximal number of identities $N_{id}^{strict}$, which satisfy the constraint $d_0^{\mathcal{E}} > d_{ict}^{\mathcal{E}}$, we can extract from a given *Langevin* ensemble. For this purpose, we devise a simple *erosion* algorithm that iteratively removes identities from the ensemble until the constraint is satisfied for all pairs of remaining identities. This naive algorithm removes the identities with the maximal number of contacts first, one by one, until no contact is left. Figure 9h shows values of $N_{id}^{strict}$ for *Langevin* ensembles of $N_{id} = 10k$ and for different values of $d_{ict}^{\mathcal{E}}$. In this case, we set the contact distance threshold slightly bigger than the threshold, $d_0^{\mathcal{E}} = 1.1 \, d_{ict}^{\mathcal{E}}$, and plot $N_{id}^{strict}$ against computational wall-time. For comparison, we also plot the performance of the *Reject* algorithm and see that *Langevin* with erosion yields a significant numerical advantage, at least with our implementations.

## G. Repulsion Distances Threshold and *DisCo*

We are now interested in optimizing the statistics of the synthetic datasets generated by our algorithms. In particular, we study here the influence of the repulsion distance thresholds $d_0^{\mathcal{E}}$ and $d_0^{\mathcal{W}}$, for *Langevin* and *Dispersion*, respectively. We would also like to investigate the impact of the *DisCo* scale parameters $\lambda_0$.

As explained in section 3 of the paper, the *Langevin* and *Dispersion* granular-like losses functions introduced in Eq. 14 and Eq. 20, respectively, are designed to repulse samples that are too close in the embedding and latent spaces. The *Langevin* algorithm therefore controls the *inter-class embedding pairwise distances* via the parameter $d_0^{\mathcal{E}}$ as illustrated by the histograms in Figure 10a. One clearly sees that increasing the value of $d_0^{\mathcal{E}}$ shifts the peak of the distribution toward bigger pairwise embedding distances. For the largest values showed in this plot, the peaks are near $d^{\mathcal{E}} \approx \frac{\pi}{2}$, similar to the real-world data shown in Figure 10c.

Similarly to *Langevin*, the *Dispersion* algorithm controls the *intra-class latent pairwise distances* via the parameter $d_0^{\mathcal{W}}$. Since increasingly larger latent pairwise distances tend to produce increasingly dissimilar images, we expect bigger values of $d_0^{\mathcal{W}}$ to widen the *intra-class pairwise embedding distances* distribution and shift its peak towards larger values. This is indeed what we observe in the histograms in Figure 10b, where the *pairwise embedding distances* distributions of *Dispersion*

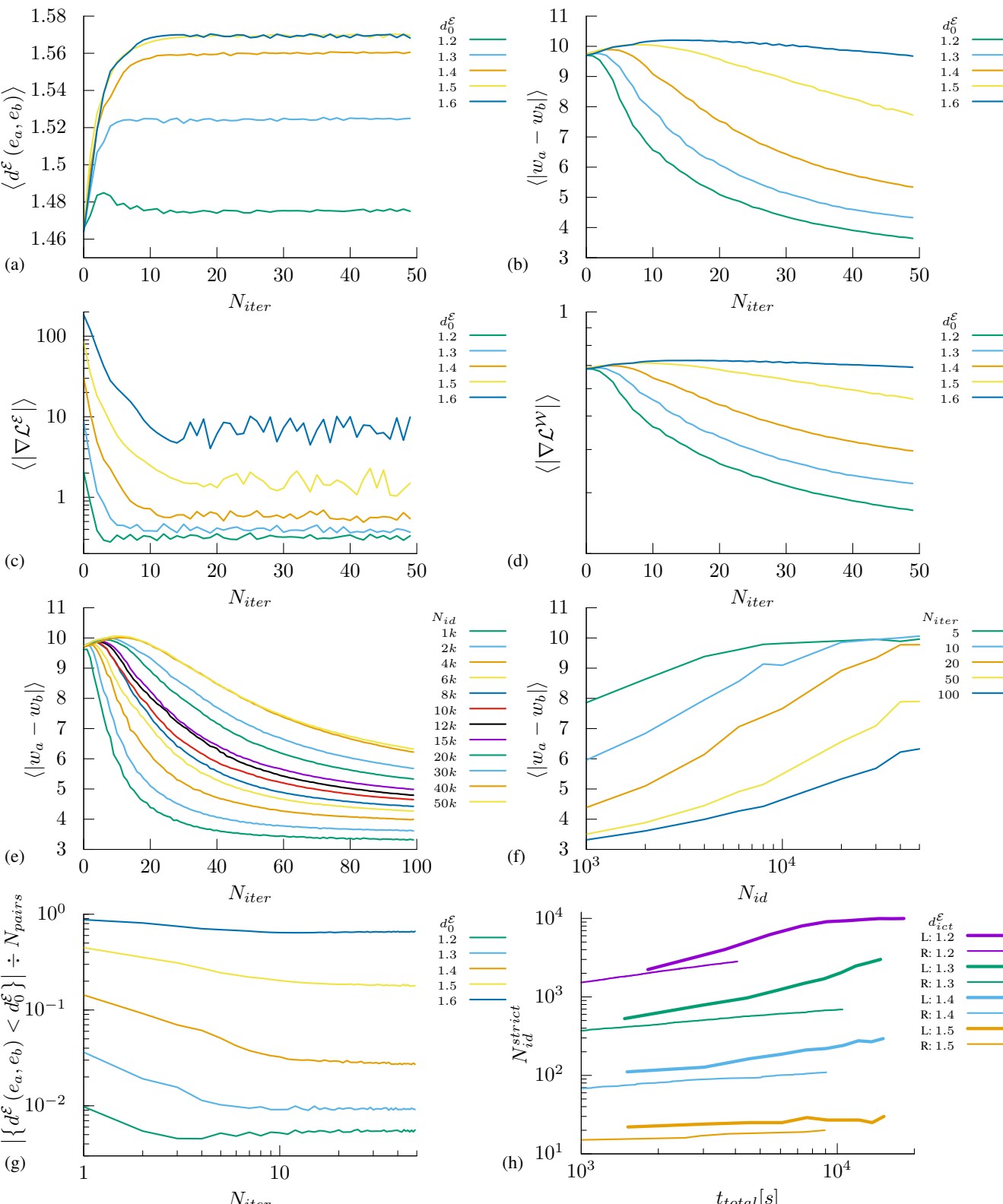

Figure 9: Dynamics of *Langevin* ensembles: Evolution of the average pairwise embedding distance a and the average pairwise latent distance b. Evolution of the average embedding contact force c and average latent pull-back force d. Average pairwise latent distance for different $N_{id}$ e and pairwise latent distance for $N_{id}$ identities after $N_{iter}$ iterations f. Proportion of pairwise embedding distances below the threshold $d_0^{\mathcal{E}}$ for $N = 10k$ identities g. Compute time used to generate $N_{id}^{strict}$ dissimilar identities for *Langevin* (L) with erosion and for *Reject* (R) and with different $d_{ict}^{\mathcal{E}}$ values h.

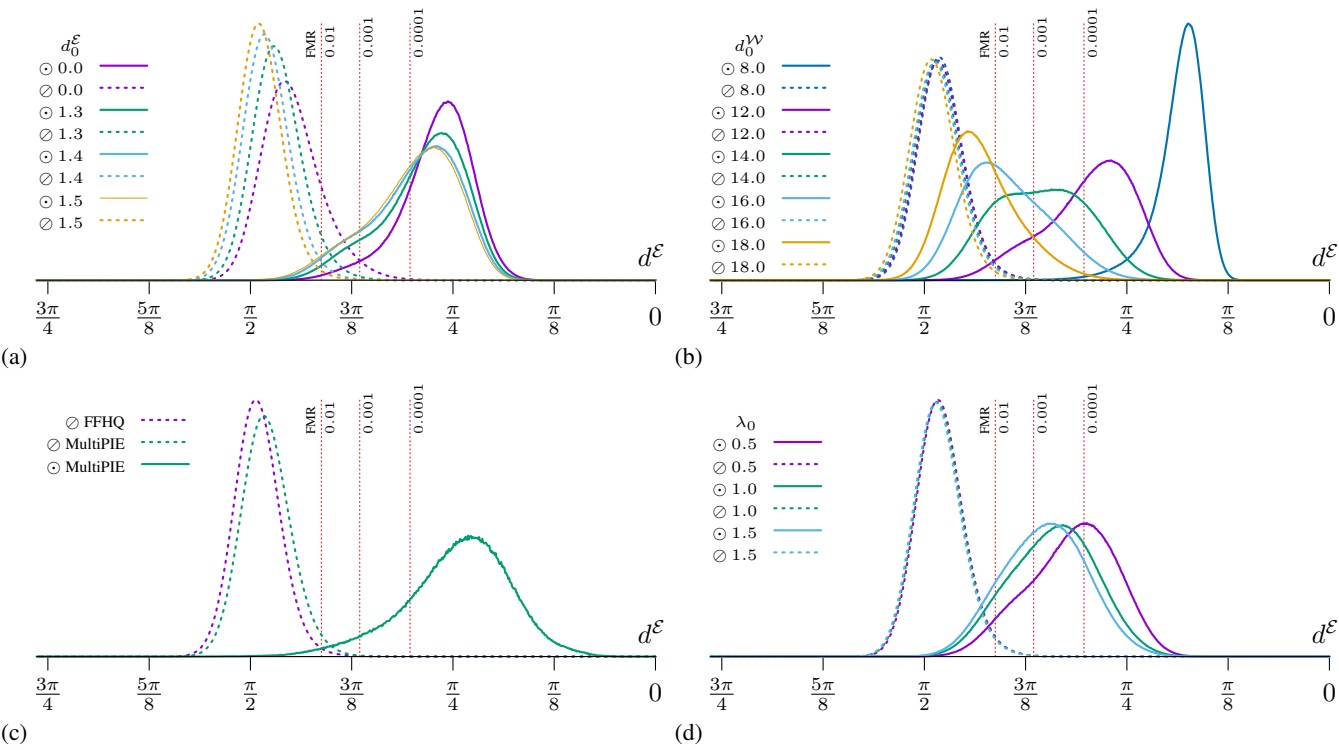

Figure 10: *intra-class* ⊙ and *inter-class* ⊘ embedding distance histograms of synthetic datasets generated with the *Langevin*, *Dispersion* and *DisCo* algorithms as well as genuine real-world data. In a, the *Langevin* repulsion distance $d_0^{\mathcal{E}}$ is shown to affect the *inter-class* distance statistics. In b and d, the *Dispersion* repulsion distance $d_0^{\mathcal{W}}$ and *DisCo* scale $\lambda_0$ are shown to affect the *intra-class* distance statistics. In c, genuine datasets histograms are shown for comparison. The vertical dashed lines, denoted "FMR", show the distances for the reference FR backbone evaluated on IJB-C dataset for three different FMR values.

Table 8: Influence of *Langevin* repulsion distance threshold $d_0^{\mathcal{E}}$ on FR accuracy.

| $d_0^{\mathcal{E}}$ | $N_{iter}$ | LFW | CPLFW | CALFW | CFP | AgeDB | Average |
|---|---|---|---|---|---|---|---|
| 1.2 | 50 | 90.82 | 59.73 | 80.97 | 60.66 | 72.65 | 72.97 |
| 1.3 | 50 | 92.78 | 63.32 | 83.33 | 63.51 | 74.37 | 75.46 |
| 1.4 | 50 | 94.43 | 65.08 | **85.13** | 65.6 | **76.97** | 77.44 |
| 1.5 | 50 | **94.67** | **65.98** | 85.1 | **67.83** | 75.67 | **77.85** |
| 1.6 | 50 | 92.73 | 65.15 | 82.28 | 67.3 | 73.02 | 76.10 |

Table 9: Influence of *Dispersion* repulsion distance threshold $d_0^{\mathcal{W}}$ on FR accuracy.

| $d_0^{\mathcal{W}}$ | $N_{iter}$ | LFW | CPLFW | CALFW | CFP | AgeDB | Average |
|---|---|---|---|---|---|---|---|
| 8.0 | 20 | 84 | 59.57 | 70.62 | 59.99 | 63.57 | 67.55 |
| 12.0 | 20 | 94.43 | 65.08 | 85.13 | 65.6 | 76.97 | 77.44 |
| 14.0 | 20 | **96.48** | **65.62** | **89.53** | **67.9** | **83.93** | **80.69** |
| 16.0 | 20 | 96.15 | 62.47 | 88.43 | 67.54 | 82.83 | 79.48 |
| 18.0 | 20 | 95.25 | 63.3 | 87.12 | 66.41 | 81.2 | 78.66 |
| 20.0 | 20 | 54.33 | 50.07 | 49.83 | 55.04 | 51.68 | 52.19 |

ensembles, created with a wide range of value of the parameter $d_0^{\mathcal{W}}$, are shown. We observe that a small value of $d_0^{\mathcal{W}} = 8.0$ lead to a very narrow distribution, while bigger values greatly shift the peak towards larger $d^{\mathcal{E}}$ values.

We are interested on the accuracy of the FR models trained on such synthetic ensembles, for different values of these two parameters. Table 8 and Table 9 show such a survey, for five different values of $d_0^{\mathcal{E}}$ and six different values of $d_0^{\mathcal{W}}$. As we can see from the first table, the values $d_0^{\mathcal{E}} = 1.4$ and $d_0^{\mathcal{E}} = 1.5$ of the *Langevin* repulsion distance threshold yield the best FR performance, while further increasing this parameter to $d_0^{\mathcal{E}} = 1.6$ degrades the overall performance. As we can see by comparing peaks of pairwise *inter-class* distances histograms of genuine and synthetic data in Figure 10, these best performing parameters values yield distributions close to genuine data from the *FFHQ* and *MultiPIE* datasets. Keeping the fixed value $d_0^{\mathcal{E}} = 1.4$, we see from the second table that the best performance is achieved with the parameter value $d_0^{\mathcal{W}} = 14.0$. It is interesting to note that the values $d_0^{\mathcal{W}} = 12.0$, $d_0^{\mathcal{W}} = 16.0$ and $d_0^{\mathcal{W}} = 18.0$ yields almost similar performance figures, but with widely different pairwise distances distributions.

Table 10: Influence of *DisCo* parameter $\lambda_0$ on FR accuracy.

| $d_0^{\mathcal{W}}$ | $\lambda_0$ | LFW | CPLFW | CALFW | CFP | AgeDB | Average |
|---|---|---|---|---|---|---|---|
| 12.0 | 0.0 | 94.43 | 65.08 | 85.13 | 65.6 | 76.97 | 77.44 |
| 12.0 | 0.5 | 95.65 | 69.6 | 87.45 | 68.54 | 80.32 | 80.31 |
| 12.0 | 1.5 | 96.2 | 73.25 | 87.7 | 73.89 | 80.73 | 82.35 |
| 12.0 | 1.0 | 96.6 | 74.77 | 87.77 | 73.89 | 80.7 | 82.75 |
| 14.0 | 1.5 | **97.22** | **75.85** | **89.48** | **78.76** | **83.32** | **84.93** |
| 16.0 | 2.0 | 94.82 | 65.18 | 85.00 | 72.39 | 77.27 | 78.93 |

In section 3 of the paper, we introduced a modification of *Dispersion*, the *DisCo* algorithm, that initialize the *intra-class* variations with a random linear combination of the *covariates* vectors introduced in (Colbois et al., 2021). This modification is introduced to further increase the *intra-class* variability of the final datasets and the implementation introduces an additional parameter $\lambda_0$ that controls the scale of the random weights $\lambda_{aI}^{\alpha} \in [-\lambda_0, \lambda_0]$. Table 10 shows the performance of five such ensembles compared to the pure *Dispersion* baseline $\lambda_0 = 0$. The first three datasets use a latent repulsion distance of $d_0^{\mathcal{W}} = 12.0$ while the fourth and last further increase this parameter. As demonstrated by the results in this table, this algorithm yields significant performance improvement compared to the original one using purely random initialization, at least up to a value of $\lambda_0 = 1.5$. This gain is of between two and four percents for the LFW, CALFW and AgeDB benchmarks, which is already quite significant. Where the *DisCo* yields the biggest advantage however is on the CPLFW and CFP benchmarks, where improvements of up to twelve percents are observed. This indicates that the *DisCo* algorithm helps generates more diversity, and that it is helpful in mitigating the relatively poor performances of StyleGAN2 ensembles on pose benchmarks.

Table 11: Influence of training set repulsion on FR accuracy.

| $d_{tr}^{\mathcal{E}}$ | $d_0^{\mathcal{E}}$ | LFW | CPLFW | CALFW | CFP | AgeDB | Average |
|---|---|---|---|---|---|---|---|
| 0.0 | 1.4 | 94.43 | 65.08 | 85.13 | 65.6 | 76.97 | 77.44 |
| 0.6 | 1.4 | **94.97** | 65.15 | 85.62 | 65.19 | 78.42 | 77.87 |
| 0.8 | 1.4 | 94.95 | **66.28** | 86.5 | 65.24 | 78.9 | 78.37 |
| 1.0 | 1.4 | 94.77 | 65.23 | 86.22 | 66.61 | 78.33 | 78.23 |
| 1.2 | 1.4 | 94.95 | 65.08 | 85.97 | 65.56 | 79.07 | 78.13 |
| 1.3 | 1.4 | 94.58 | 65.18 | **86.37** | 65.91 | **79.85** | **78.38** |
| 1.4 | 1.4 | 94.08 | 65.62 | 85.37 | **66.93** | 77.12 | 77.82 |
| 1.5 | 1.4 | 91.85 | 64.58 | 81.58 | 66.37 | 71.72 | 75.22 |

# H. Identity Leakage from Generator Training Set and Mitigation

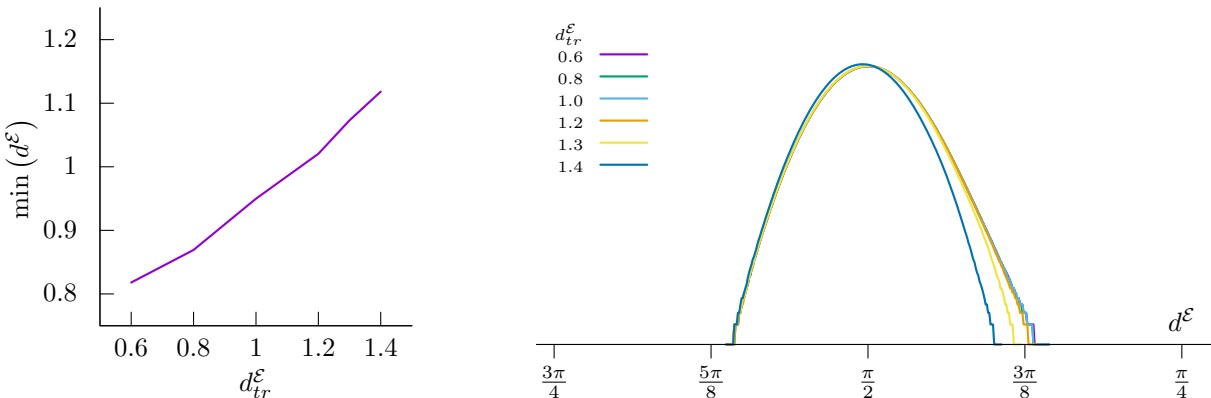

Figure 11: Effect of the $d_{tr}^{\mathcal{E}}$ parameter on synthetic to training set distances distribution: On the left, effect on the minimum value, on the right: effect on the distribution tail.

While we have generated a large quantity of synthetic data, it is important to emphasis that the generator is trained on genuine data. We want to verify that the synthetic data generated by our algorithms is sufficiently far from the original training set data and, if it is not the case, find appropriate mitigation techniques to control potential *training set leakage*. It has been shown that, despite the care taken by the authors of previous methods such as DCFace (Kim et al., 2023), their method still generates a significant number of samples that are very close, if not indistinguishable, from images from the generator training set (Shahreza & Marcel, 2024).

Since GANs-based generators are less prone to leakage than diffusion-based models (Carlini et al., 2023), we developed our method based on GAN models. Another advantage of our method is that we can further mitigate the problem by introducing an additional interaction that repulses the synthetic samples away from the training set samples. This is quite easily implemented in *Langevin* by adding a loss function similar to Eq. 14 that repulse synthetic samples away from generator training set samples embeddings

$$\mathcal{L}_{tr}^{\mathcal{E}} = \frac{k_{tr}^{\mathcal{E}}}{2} \sum_{a=1}^{N_{id}} \sum_{A=1}^{N_{tr}} \begin{cases} \left(d_{tr}^{\mathcal{E}} - d_{aA}^{\mathcal{E}}\right)^2 & : d_{aA}^{\mathcal{E}} < d_{tr}^{\mathcal{E}} \\ 0 & : d_{aA}^{\mathcal{E}} > d_{tr}^{\mathcal{E}} \end{cases} \tag{29}$$
$$d_{aA}^{\mathcal{E}} = d^{\mathcal{E}}\left(e\left(w_a\right), E_A\right),$$

where $N_{tr}$ is the number of training set samples, where $A = 1 \dots N_{tr}$ and where $E_A$ is the embedding of the $A$-th sample. The total loss function for the *Langevin* algorithm Eq. 18 changes to

$$\mathcal{L}^{(t)} = \mathcal{L}^{\mathcal{E}} + \mathcal{L}^{\mathcal{W}} + \mathcal{L}_{tr}^{\mathcal{E}}. \tag{30}$$

We introduced a new parameter $d_{tr}^{\mathcal{E}}$ which is similar to the pairwise embedding repulsion distance $d_0^{\mathcal{E}}$ but for granular interactions between synthetic and genuine samples. This parameter has the effect of shifting the tail of the genuine-synthetic

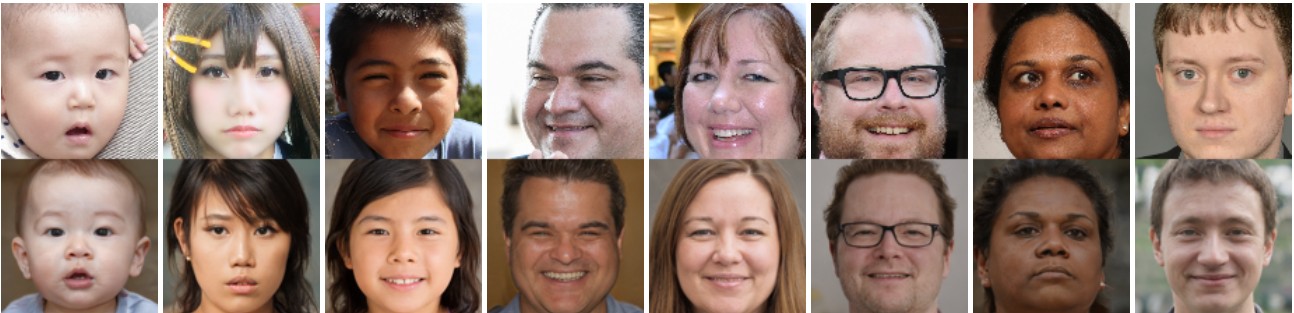

Figure 12: Top row: Selection of the most similar images among the 64 nearest embedding pairs between the original FFHQ (top) and images generated with *Langevin* algorithm with $d_{tr}^{\mathcal{E}} = 0.6$ and $N_{id} = 10k$ (bottom). Note that (similar to the left image) there are several images of children among similar images, which are hard to compare and therefore we focused on adults for this study.

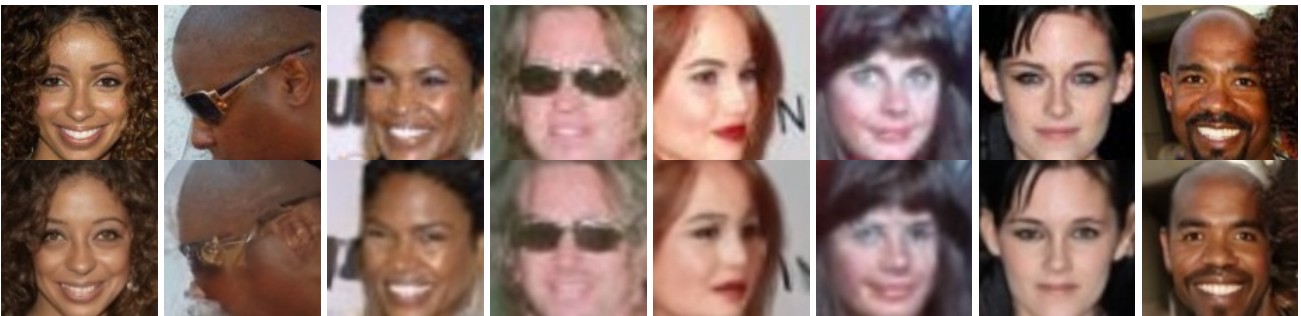

Figure 13: Top row: Selection of the most similar images among the 64 nearest embedding pairs between the original CASIA-WebFace (top) and synthetic images in the **DCFace** dataset (bottom).

embedding distance histogram, as shown in Figure 11 thus controlling the most problematic images in term of embedding distance. Table 11 reports the effect of this new parameter on FR accuracy, for several values of $d_{tr}^{\mathcal{E}}$. We observe that, surprisingly, increasing the value $d_{tr}^{\mathcal{E}}$ slightly increases FR accuracy, and the biggest value considered $d_{tr}^{\mathcal{E}} = 1.3$ performs slightly better than no training set repulsion at all.

To verify the leakage of identity in the generated datasets, similar to (Shahreza & Marcel, 2024), we compare all images in the synthetic dataset with all images in the training set of generator model. Figure 12 and Figure 13 illustrate the most similar pairs of genuine and synthetic images with the smallest embedding distance for both our method and DCFace, respectively. While it is difficult to conclude the leakage in our synthetic dataset, we can see that DCFace has almost memorized several samples, indicating a serious identity leakage in the DCFace dataset.

## I. Synthetic Dataset Scaling and Face Recognition Performance

Having performed a detailed analysis of the *Langevin*, *Dispersion* and *DisCo* algorithms as well as the influence of their respective parameters, we would like to also see if we can scale our approach to bigger synthetic datasets. Table 12 shows the result of this survey. In this table, we also show the effect of compounded datasets, datasets that have the same biometric references, created with *Langevin*, but different variations. In particular, several datasets are compounded with their reference *Langevin* ensemble, thus adding one variation per class, denoted by a check-mark in the Ref. column of the table. As we can see, this offers a very limited performance increase but we decide to keep this for bigger datasets as the data has to be computed anyway.

Scaling up to $N_{id} = 30k$ however offers an impressive performance improvement, in particular with *DisCo* variations, with an impressive average score of 88.40, almost reaching the performance of diffusion based methods. For $N_{id} = 30k$ we also study the influence of the number of variations, showing that $N_{var} = 64$ yields the best results. Surprisingly, for $N_{var} = 128$ the performance drops very significantly, by almost twenty points. This might be explained by the lack of

Table 12: Influence of the number of identities, number of variations and variation-creation method on FR accuracy.

| Method | $N_{id}$ | $N_{img}$ | Ref.† | $N_{var}$ | $d_0^{\mathcal{W}}$ | $\lambda_0$ | LFW | CPLFW | CALFW | CFP | AgeDB | Average |
|---|---|---|---|---|---|---|---|---|---|---|---|---|
| *Langevin + Dispersion* | 10'000 | 640'000 | - | 64 | 12.0 | - | 94.43 | 65.08 | 85.13 | 65.6 | 76.97 | 77.44 |
| | | 650'000 | ✓ | 64 + 1 | 12.0 | - | 94.38 | 65.75 | 86.03 | 65.51 | 77.3 | 77.79 |
| | 30'000 | 480'000 | - | 16 | 12.0 | - | 90.65 | 64.42 | 80.12 | 63.07 | 70.18 | 73.69 |
| | | 960'000 | - | 32 | 12.0 | - | 94.83 | 66.25 | 84.3 | 65.51 | 76.1 | 77.40 |
| | | 1'620'000 | - | 64 | 12.0 | - | 97.98 | 71.57 | 91.7 | 72.76 | 88.87 | 84.58 |
| *Langevin + DisCo* | 10'000 | 640'000 | - | 64 | 12.0 | - | 96.6 | 74.77 | 87.77 | 73.89 | 80.7 | 82.75 |
| | 20'000 | 1'300'000 | ✓ | 64 + 1 | 12.0 | 1.5 | 98.22 | 77.57 | 91.93 | 78.93 | 89.95 | 87.32 |
| | | 1'300'000 | ✓ | 64 + 1 | 14.0 | 1.5 | 98.53 | 81.17 | 92.93 | 83.56 | 92.32 | 89.70 |
| | | 1'950'000 | ✓ | 64 + 1 | 12.0 | 1.5 | 98.37 | 79.53 | 92.58 | 80.71 | 90.82 | 88.40 |
| | | 3'870'000 | ✓ | 128 + 1 | 12.0 | 1.5 | 86.15 | 61.98 | 70.35 | 68.23 | 59.87 | 69.32 |
| | 30'000 | 1'470'000 | ✓ | 48 + 1 | 14.0 | 1.5 | 98.55 | 81.32 | 93.33 | 83.37 | 92.05 | 89.72 |
| | | 1'950'000 | ✓ | 64 + 1 | 14.0 | 1.5 | **98.97** | **81.52** | **93.95** | **83.77** | **93.32** | **90.31** |
| | 40'000 | 2'600'000 | ✓ | 64 + 1 | 12.0 | 1.5 | 98.33 | 78.25 | 92.27 | 79.31 | 89.43 | 87.52 |
| | 50'000 | 3'250'000 | ✓ | 64 + 1 | 12.0 | 1.5 | 98.47 | 79.88 | 92.57 | 79.93 | 90.33 | 88.24 |
| *Langevin + Covariates* | 10'000 | 180'000 | ✓ | 17 + 1 | - | - | 77.68 | 59.2 | 64.07 | 61.07 | 52.55 | 62.91 |

† References created with *Langevin* are compounded with the variations for FR model training.

Table 13: Bias evaluation of trained face recognition models on the Racial Faces in-the-Wild (RFW) dataset

| Dataset | Type | Caucasian | Asian | Indian | African | Avg. | Std. |
|---|---|---|---|---|---|---|---|
| SynFace (Qiu et al., 2021) | GAN | 65.60 | 64.48 | 61.48 | 57.27 | 62.21 | 3.01 |
| SFace (Boutros et al., 2022) | GAN | 75.68 | 69.7 | 70.63 | 66.23 | 70.56 | 2.08 |
| IDNet (Kolf et al., 2023) | GAN | 70.03 | 64.22 | 65.77 | 59.3 | 64.83 | 2.89 |
| GANDiffFace (Melzi et al., 2023) | GAN | 76.32 | 72.85 | 72.45 | 66.48 | 72.03 | 3.00 |
| ExFaceGAN (Boutros et al., 2023b) | GAN | 65.25 | 65.40 | 64.25 | 57.97 | 63.22 | 3.28 |
| DigiFace (Bae et al., 2023) | Computer Graphics | 71.93 | 68.30 | 69.02 | 64.8 | 68.51 | 1.93 |
| IDiff-Face (Uniform) (Boutros et al., 2023a) | Diffusion | 84.92 | 80.63 | 81.67 | 75.65 | 80.72 | 2.72 |
| DCFace-1.2M (Kim et al., 2023) | Diffusion | 90.08 | 82.97 | 87.07 | 80.97 | 85.27 | 2.66 |
| ***Langevin-DisCo*-1.6M [ours]** | GAN | 89.17 | 82.95 | 85.43 | 81.42 | 84.74 | 1.81 |

*inter-class latent repulsion*, a design choice made for simplicity, as explained previously.

## J. Bias Evaluation

To investigate the bias in our synthetic dataset, we evaluate the performance of face recognition models trained with our dataset and compare with previous datasets on the Racial Faces in-the-Wild (RFW) (Wang et al., 2019) dataset. As the results in Table 13 show, our method has achieved the best performance compared to GAN-based synthetic datasets and comparable average performance with diffusion-based methods. In terms of bias, this table demonstrates that our method achieves the lowest standard deviation for recognition accuracy across different demographic groups, indicating the lowest bias in the trained face recognition model using our dataset.

## K. Closing the Gap with Real-World Data

To better assess the performance of our best datasets, Figure 14 shows the Receiver Operating Characteristic (ROC) curves for a number of models trained on synthetic and genuine data of the IARPA Janus Benchmark-B (IJB-B) (Whitelam et al., 2017) and IARPA Janus Benchmark-B (IJB-C) (Maze et al., 2018) datasets. As can be observed in this figure, our method outperforms all GAN-based synthetic datasets in the literature. In addition, our method achieves comparable performance with diffusion-based datasets for high values of FMR; however, for low values of FMR, diffusion-based methods achieve better performance than our method. We should note that, as mentioned in section 2 of the paper, DCFace is generated with a dual condition model trained on CASIA-WebFace (Yi et al., 2014) with identity labels, and therefore the generator model

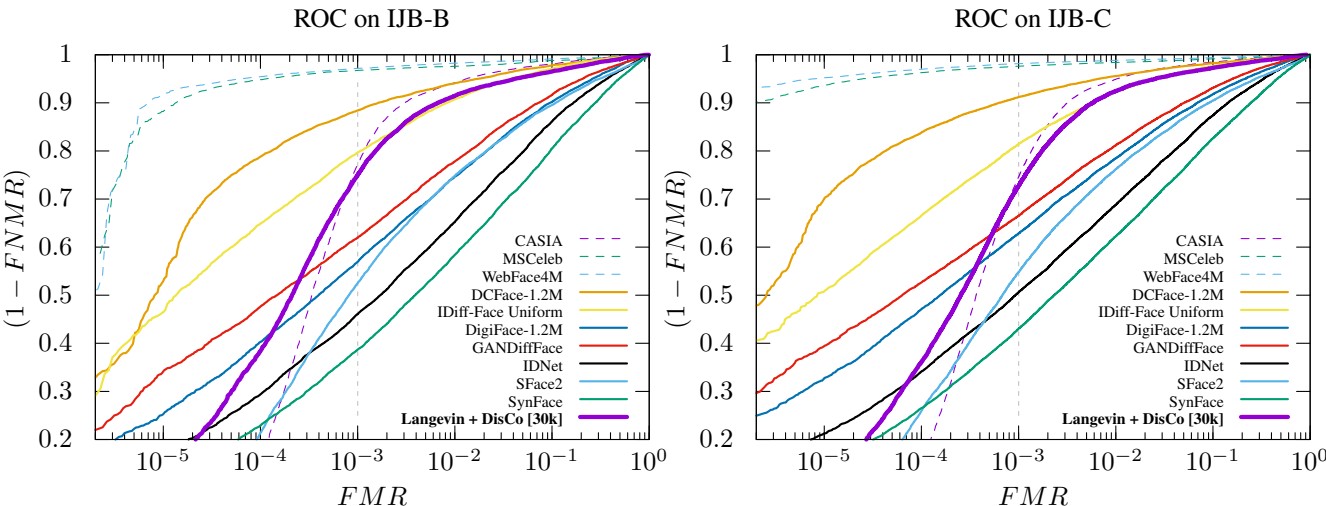

Figure 14: ROC curves for models trained on synthetic datasets, benchmarked on the IJB-B and IJB-C datasets.

leverages[1] the identity information in the CASIA-WebFace dataset (Shahreza & Marcel, 2024). In contrast, our method is based on a pretrained generative model, which is trained on unlabeled face images from the FFHQ dataset. In addition, as discussed in Appendix H, there is identity leakage in the DCFace dataset. Compared to FR models trained with real data, the ROC curves show that there is still a gap between training with real and synthetic datasets. Nevertheless, the promising improvement achieved by our method for GAN-based models reveals potential in generative models and training FR with synthetic data, which require further research in the future.

## L. Future Directions

While we showed that our algorithms can yield very good synthetic face datasets, many improvements are still to be made. Most importantly, our method crucially relies on a reference off-the-shelf FR model. In some sense, the datasets generated with our method iteratively *learns*, by stochastic gradient descent, from the reference FR model, and therefore quite certainly learn its biases as well. A possible fix that could mitigate, at least partially, this fundamental issue is to use several reference models in parallel, simply by adding more embedding losses to the algorithms.

Fundamentally, the problem of generating complex synthetic data, that has similar characteristics that some genuine reference, is a *chicken-and-egg* problem. It could be perhaps constructive to reformulate this problem in a way that makes it clear that the synthetic data will always depends to some extent on genuine priors, and rather insist on developing procedures giving guaranties that the original data is at least hard to recover and that the synthetic dataset gives a good enough representation of reality. A few steps in this direction were performed in this appendix, possibly also hinting at some interesting research directions toward a much deeper problem: generalization of deep neural networks.

We would like to stress that, while large part of this work was devoted to optimization of these algorithms, they remain computationally expensive. For each sample an image must be generated and its embedding computed, twice per iteration with one backward pass, which is computationally expensive as the data goes through an intermediary image space of very high dimensionality. On the other hand the only function we are interested in is an identity aware metric $d_{id}^{\mathcal{W}}$ on the latent space of a given generator. It is certainly possible to learn such a function to a relevant degree of accuracy and to use this instead of the full chain, at least for bootstrapping. Alternatively, while technically challenging, it is certainly possible to prune connections and weights in the $\mathcal{W} \rightarrow \mathcal{E}$ generator-feature-extractor chain, which would vastly improve efficiency.

---

[1]Since DCFace is trained on CASIA-WebFace dataset with identity labels, the recent Synthetic Data for Face Recognition (SDFR) Competition held in conjunction with the 18th IEEE International Conference on Automatic Face and Gesture Recognition (FG 2024) disqualified all submissions based on DCFace (Shahreza et al., 2024). It is also shown in (Shahreza & Marcel, 2024) that DCFace has critical identity leakage from CASIA-WebFace dataset.

