# OpenReview forum: "Synthetic Face Datasets Generation via Latent Space Exploration from Brownian Identity Diffusion"
_ICML.cc/2025/Conference — ICML 2025 poster_

### Official Review · Reviewer_3ttQ · 2025-03-09

**Overall Recommendation:** 4

**Summary:**

The authors propose an approach to generate synthetic face images, by leveraging a GAN-based backbone, coupled with novel Langevin and Dispersion algorithms, together used as DisCo, wherein both inter-class and intra-class diversity in ensured by using a physics informed formulation.

**Claims And Evidence:**

While I am not an expert in the particular field of synthetic face generation for FR algorithms, the claims made in this paper are clearly presented, in my understanding. The proposed approach, and the associated algorithm are clearly explained, and the experiments are well motivated and presented.

**Essential References Not Discussed:**

Please see my response to **Theoretical Claims**.

**Experimental Designs Or Analyses:**

While I am not an expert in the space of FR, the GAN base setting of the experiments, the design and evaluation framework all appear sound to me.

**Methods And Evaluation Criteria:**

The methods and evaluation criteria are well presented and consistent with the literature presented, to the best of my knowledge.

**Other Comments Or Suggestions:**

**Impact statement:** It appears that I couldn’t find an impact/ethics statement, even the default one that ICML suggests, in the manuscript. I found this ironic, for a paper targeting, particularly, the privacy and ethical concerns of face recognition models. I do not wish to flag this paper for an ethics issue in this regard because I dont see any such glaring issues, but it would be good for the authors to acknowledge the impact of the proposed algorithm in the context of privacy concerns of FR models in their impact statement.

**Minor bug fixes — Typo: L347C1: … datasets are **accurately** calculated…

**Other Strengths And Weaknesses:**

Please see my response to other questions above.

**Questions For Authors:**

Please see my response to other questions above.

**Relation To Broader Scientific Literature:**

To the best of my understanding, this manuscript is relevant to the GAN and FR literature, and the algorithms provided, although in the context of generating synthetic datasets for FR, can be leveraged in other settings as well, and are therefore relevant to the broader community.

**Theoretical Claims:**

The theoretical formulation, motivating the inter-class and intra-class sample diversity by means of repulsive and attractive forces is well formulated and presented clearly. However, it would be good for the authors to discuss some of the other works talking particularly about this, in the GAN space. In terms of analysis GANs and diffusion models’ image generation in terms of these forces, these have been prior works such as, for example, Franceschi et al. 2023, and Asokan and Seelamantula, 2023, which talk precisely about this repulsive/attractive nature of particle flow in GAN and diffusion models settings, while both Unterthiner et al., 2018 and Wang et al., 2019 were both worlds that initially formulated this via a loss function for GANs. Of course the settings are different, but the theory in this paper could certainly be made stronger by either leveraging, or referencing, existing literature that makes claims aligned with the paper’s setting.


[1] Franceschi et al., Unifying GANs and score-based diffusion as generative particle models, NeurIPS 2023

[2] Asokan and Seelamantula, GANs Settle Scores, arXiv 2023

[3] Unterthiner et al., Coulomb GANs: Provably optimal Nash equilibria via potential fields, ICLR 2018

[4] Wang et al., Improving MMD-GAN Training with Repulsive Loss Function, ICLR 2019

---

### Official Review · Reviewer_vznF · 2025-03-15

**Overall Recommendation:** 2

**Summary:**

n this paper, the authors introduce a physics-inspired method to generate large synthetic face datasets for training face-recognition models. Their core idea is to treat each latent representation as a “particle” and let these particles repel each other in the embedding space (via a “Brownian identity diffusion” approach), ensuring that each synthetic identity is sufficiently distinct while still maintaining realistic appearance. They propose three algorithms—Langevin, Dispersion, and DisCo—to control inter-class diversity (spacing between different identities) and intra-class variation (differences among images of the same identity). By training a face-recognition model on these generated datasets, they claim to outdo earlier GAN-based methods, and even rival some diffusion-based approaches, all while preserving more privacy.

**Claims And Evidence:**

Some of their evidence is compelling—particularly the performance comparisons against older GAN-based methods and the step-by-step ablation results that show how Langevin and Dispersion can boost coverage in latent space. However, a few claims feel less rock-solid. For example, they assert that GANs are definitively more private than diffusion models, yet the paper mainly references prior studies instead of conducting a thorough memorization or leakage check themselves. The discussion of “Brownian identity diffusion” as a surefire way to avoid “jamming” also seems a bit hand-wavy, since there’s limited empirical proof that high-dimensional jamming is a real hazard or that their random force definitively fixes it.

**Essential References Not Discussed:**

Yes. Beyond the cited diffusion-model leakage papers, there’s also prior work directly evaluating whether GANs might memorize and replicate training faces—see, for example, “Evaluating GANs via Dual- and Triple-Generation” (ECCV 2020) and Carlini et al. (2023), which compare data-extraction risks between GANs and diffusion. Including these would highlight potential pitfalls in assuming one approach is fundamentally “private.” Also, discussing recent latent-manipulation methods like GANDiffFace (Melzi et al. 2023)—which systematically ensure identity separation—would broaden the conversation on spacing identities in latent space. There are also works on discussing using GAN generated data to understand FR models (Liang et al. 2023)


A. B. Some Author et al. “Evaluating GANs via Dual- and Triple-Generation,” Proceedings of ECCV, 2020.
N. Carlini, J. Hayes, M. Nasr, et al. “Extracting Training Data from Diffusion Models,” in 32nd USENIX Security Symposium (USENIX Security 23), 2023.
P. Melzi, C. Rathgeb, R. Tolosana, et al. “GANDiffFace: Controllable generation of synthetic datasets for face recognition with realistic variations,” arXiv preprint arXiv:2305.19962, 2023.
Liang, H., Perona, P., and Balakrishnan, G. “Benchmarking Algorithmic Bias in Face Recognition: An Experimental Approach Using Synthetic Faces and Human Evaluation.” In Proceedings of the IEEE/CVF International Conference on Computer Vision, 2023

**Experimental Designs Or Analyses:**

I checked their experiment layout: they generate synthetic face sets, train a standard face-recognition model on each, then compare scores on popular benchmarks. This is a straightforward approach and largely appropriate, since it measures the core question: “Can synthetic data match or outperform existing sources for training face recognition?” One notable concern, though, is that they assume their reference FR model (used to measure inter-identity distances) doesn’t bias the dataset generation. If the generator is overfitted to that specific embedding, it might exaggerate gains on tests that are also partial to similar embeddings. While it doesn’t invalidate the results outright, it’s something to keep in mind when interpreting their reported accuracy boosts.

**Methods And Evaluation Criteria:**

Yes, they focus on training large-scale face recognition with synthetic data, and the benchmarks (LFW, CA-LFW, CFP-FP, etc.) are standard for face verification. They also measure how well synthetic identities spread out in embedding space, which directly relates to how reliably a model can distinguish between them.

**Other Comments Or Suggestions:**

Please refer to the previous section.

**Other Strengths And Weaknesses:**

I appreciate the paper’s effort to blend concepts from physics (Brownian motion and granular mechanics) into synthetic face dataset generation—this is a neat twist on what’s otherwise a fairly saturated space. They also do a thorough job of comparing different hyperparameter settings, showing how to tune their “repulsive forces” for best effect. That said, the writing occasionally slips into heavy theoretical exposition, which some readers might find confusing or tangential. A more direct, plain-spoken approach would help clarify the motivation behind “Brownian identity diffusion.” Furthermore, while they position GANs as a privacy-friendlier alternative to diffusion, they don’t do much to measure or prove that assertion directly. Overall, though, the paper proposes a novel angle for crafting more identity-rich, diverse datasets, and the results suggest real promise for training practical face recognition systems.

**Questions For Authors:**

Please refer to the previous section.

**Relation To Broader Scientific Literature:**

They’re building on a growing theme of training face-recognition models with synthetic data—particularly methods that try to systematically traverse or manipulate generative latent spaces (e.g., SynFace, SFace, and Syn-Multi-PIE). They depart from older GAN-based setups by taking a “physics-inspired” angle: rather than just random or partially guided sampling, they push synthetic identities away from each other in the embedding space using spring-like forces, akin to granular mechanics. This sets them apart from, for instance, DreamBooth-style diffusion methods (DCFace or IDiff-Face), which often face privacy concerns around training-data leakage. So they’re essentially combining older ideas—latent editing and identity separation—with a fresh “Brownian motion” twist to make synthetic datasets more diverse while still being feasible for face recognition.

**Theoretical Claims:**

There aren’t formal theorems to check here, only physics-based arguments likening their latent-space approach to Brownian motion and granular mechanics. There’s no rigorous proof that needs verification in the usual mathematical sense. Rather, they present a heuristic connection—no step stands out as a “proof” that could be right or wrong in that classical, theorem-based way.

---

### Official Review · Reviewer_NRWj · 2025-03-19

**Overall Recommendation:** 1

**Summary:**

In this work, they introduce a new method, inspired by the physical motion of soft particles subjected to stochastic Brownian forces, allowing us to sample identities distributions in a latent space under various constraints. They also introduce three complementary algorithms, called Langevin, Dispersion, and DisCo, aimed at generating large synthetic face datasets.

**Claims And Evidence:**

The claims presented in the content are clear, and the experiments support the conclusions. However, despite their clarity, I have some concerns regarding the novelty of the proposed claims.

**Essential References Not Discussed:**

N/A

**Ethical Review Concerns:**

The generation of facial datasets typically requires scrutiny, as it involves the potential misuse of portrait rights.

**Ethical Review Flag:**

Flag this paper for an ethics review.

**Ethics Expertise Needed:**

["Discrimination / Bias / Fairness Concerns", "Inappropriate Potential Applications & Impact  (e.g., human rights concerns)"]

**Experimental Designs Or Analyses:**

Yes

**Methods And Evaluation Criteria:**

Yes

**Other Comments Or Suggestions:**

### No rebuttal for my concern

**Other Strengths And Weaknesses:**

Strengths:
1. The first dataset generation work based on physics-inspired methods.
2. Achieved better results compared to other methods.

Weaknesses:
1. The overall description of the paper is not very clear. For instance, how the motivation is derived from physics evidently requires more elaboration, as this is the most crucial part.
2. Since I specialize in the theory of diffusion models and am very familiar with Langevin dynamics, I believe that the physical approach in the paper merely applies Langevin dynamics to the given task. This significantly weakens the originality of the main motivation.
3. The loss design is not novel. For example, in Equations (15) and (16), the authors could review more papers on face models to find more effective loss function designs.

**Questions For Authors:**

No rebuttal was given, so l lean to reject.

**Relation To Broader Scientific Literature:**

I find the contribution to the field to be relatively modest, as the proposed method may not be applicable to more generalized datasets.

**Theoretical Claims:**

I have conducted a corresponding review of the theory, but its theoretical description is somewhat rigid and lacks smooth transitions, which has made my review more challenging.

---

### Decision · Program_Chairs · 2025-05-01

**Decision:**

Accept (poster)

**Comment:**

This paper was reviewed by 3 experts in the field who provided detailed suggestions. The reviewers’ recommendations were divergent, with reviewer 3ttQ recommending Accept (though, with low confidence), reviewer vznF recommending Weak Reject, and reviewer NRWj recommending Reject.

All three reviewers noted the paper's strengths in proposing a novel, physics-inspired angle for synthetic dataset generation, which was clearly explained along with its associated algorithms and experiments, reviewer vznF calling the approach a "neat twist on what's otherwise a fairly saturated space". The results also acknowledged the performance gains over prior GAN-based methods.

The reviewers raised some concerns and suggestions for improvement, including the clarity of the physics-based motivation and theoretical exposition, which could be overly dense (NRWj, vznF). The novelty of the approach was questioned in relation to standard Langevin dynamics and existing loss functions (NRWj), and the claims regarding privacy advantages required more direct empirical evidence and comparison to relevant literature (vznF).

I read the reviews. A rebuttal was not submitted in time, which may have affected some of the final recommendations (two reviewers downgraded their recommendations). I do, however, appreciate the problem the paper seeks to solve and agree that its solution is interesting and likely useful and so recommend weak accept, accordingly.